# Endogenous transcripts control miRNA levels and activity in mammalian cells by target-directed miRNA degradation

Francesco Ghini[1], Carmela Rubolino[1], Montserrat Climent[1], Ines Simeone[1], Matteo J. Marzi[1] & Francesco Nicassio [1]

Little is known about miRNA decay. A target-directed miRNA degradation mechanism (TDMD) has been suggested, but further investigation on endogenous targets is necessary. Here, we identify hundreds of targets eligible for TDMD and show that an endogenous RNA (Serpine1) controls the degradation of two miRNAs (miR-30b-5p and miR-30c-5p) in mouse fibroblasts. In our study, TDMD occurs when the target is expressed at relatively low levels, similar in range to those of its miRNAs (100–200 copies per cell), and becomes more effective at high target:miRNA ratios (>10:1). We employ CRISPR/Cas9 to delete the miR-30 responsive element within Serpine1 3'UTR and interfere with TDMD. TDMD suppression increases miR-30b/c levels and boosts their activity towards other targets, modulating gene expression and cellular phenotypes (i.e., cell cycle re-entry and apoptosis). In conclusion, a sophisticated regulatory layer of miRNA and gene expression mediated by specific endogenous targets exists in mammalian cells.

[1] Center for Genomic Science of IIT@SEMM, Istituto Italiano di Tecnologia (IIT)—Via Adamello 16, 20139 Milan, Italy. These authors contributed equally: Francesco Ghini, Carmela Rubolino. These authors jointly supervised this work: Matteo J. Marzi, Francesco Nicassio. Correspondence and requests for materials should be addressed to M.J.M. (email: matteo.marzi@iit.it) or to F.N. (email: francesco.nicassio@iit.it)

MicroRNAs (miRNAs) are an evolutionarily conserved class of small (about 18–22 nt long) non-coding RNAs that function in post-transcriptional regulation of gene expression[1]. Targets are bound through base paring between the miRNA and their miRNA responsive elements (MREs), usually located in the 3′ untranslated region (3′UTR)[2]. To act as such, any MRE usually presents complementarity to bases 2–7 (the seed) of miRNAs; however, other sequences, usually located near the miRNA 3′ end, may also form additional base pairs and thus participate in target recognition. Due to the low levels of complementarity between miRNAs and their RNA targets, from hundreds to thousands RNAs could interact with the same miRNA sequence, as demonstrated by high-throughput experimental studies[3,4]. For the interaction with their targets to take place, miRNAs must be loaded onto Argonaute proteins (AGO) and form the core of the RNA-induced silencing complex (RISC). Within RISC, miRNAs induce silencing by target destabilisation and/or translational repression[5,6]. Computational methods, such as TargetScan[7] and others[8], are able to predict miRNA targets and their MREs based on seed type hierarchy (8-mer > 7-mer–m8 > 7-mer–A1 > 6-mer) and on sequence conservation of orthologous mRNAs as found by comparative genome analysis. Usually, target expression changes slightly when miRNA levels are perturbed[9,10]; however, the resulting phenotypic effect can be profound as targets often converge towards the same pathway or biological process.

Intriguingly, target:miRNA interactions have been suggested to act as a bidirectional control mechanism, with targets in turn affecting miRNAs activity. Two mechanisms have been reported: the competing endogenous RNA (ceRNA) hypothesis[11] and the target-directed miRNA degradation (TDMD) mechanism[12]. The ceRNA theory postulates that endogenous targets compete with each other for binding to a shared miRNA; therefore, a sudden change in the expression of a competing endogenous target (e.g., the ceRNA) might influence miRNA activity on other targets[13]. Most evidence in favour of the ceRNA hypothesis comes from over-expression approaches, so that the impact of ceRNAs on miRNA-mediated mechanisms in physiological settings is still debated[14–16]. In the TDMD mechanism, the RNA target (the TDMD target) promotes degradation of its miRNA[17,18], accompanied by post-transcriptional modification of the miRNA sequence, i.e., tailing (addition of nucleotides at the 3′ end) and trimming (shortening)[19], and unloading from AGO[20]. Studies performed using artificial targets showed that extended complementarity to miRNAs 3′ regions combined with a central bulge of ≤ 5 nt, promotes miRNA degradation[18,21]. However, TDMD molecular basis and physiological role are still obscure. Endogenous RNA targets implicated in TDMD and the role they play in modulating miRNA activity need to be further investigated, especially in non-neuronal cells. So far, the evidence for accelerated miRNA decay comes from studies on viral targets (e.g., the non-coding HSUR RNA and m169 mRNA[22,23]) and on artificial transcripts, both characterised either by a central bulge or by perfect complementarity[15,24]. Indeed, it has been shown that, in physiological conditions, miRNA decay can be accelerated by a rapid change in gene expression (e.g., light–dark transition or growth factor stimulation[25,26]), suggesting the existence of a post-transcriptional mechanism able to control miRNA levels. However, precise molecular details remain obscure. We and others have recently shed light on the dynamics of miRNA decay in mammalian cells by using new tailored approaches based on in vivo RNA labelling[27,28]. In our study, different pools of miRNAs were identified on the basis of their decay pattern: "slow" miRNAs, very stable ($T_{1/2} > 24$ h), and apparently downregulated through dilution by cell division; and "fast" miRNAs, quickly turned over in the cell ($T_{1/2}$ from 4 to 14 h). Intriguingly, we found a significant association between miRNA type of decay ("fast" or "slow") and number and expression levels of targets bearing an additional complementarity to miRNA 3′ ends (3C-targets[27]), as required for TDMD. On these bases, we hypothesised that TDMD is a general mechanism, with specific endogenous targets able to control the dynamics of degradation of their miRNAs.

In this study, we provide evidence that the endogenous target Serpine1 is able to control the levels of the two miRNAs miR-30b-5p and -30c-5p by inducing their degradation during cell cycle re-entry of quiescent fibroblasts stimulated with serum. Following an in-depth investigation of the interaction between Serpine1 and miR-30b/c, we describe TDMD as a post-transcriptional mechanism that controls levels and activity of miRNAs.

## Results

**Candidate decay targets are present in the mammalian genome.** Using the TargetScan database as a source of miRNA:target pairs, we identified endogenous RNAs that could trigger miRNA degradation and focused our analysis on targets presenting a 3′ pairing contribution (3C-score[29]) on top of the canonical seed pairing (3C-targets, Fig. 1a and Supplementary Data 1). Overall, nearly 15% of all the target:miRNA predicted interactions involved a 3C site. However, most of these interactions were characterised by limited complementarity (e.g., three or four base pairs) at the miRNA 3′ end, which, according to previous studies with artificial targets[18,20,21], should not be sufficient for triggering TDMD. Based on their 3C-scores, 3C-targets were divided into three classes: (i) low, with 3C-scores between −0.01 and -0.03; (ii) mid, scoring between −0.03 and −0.05; and (iii) high, with 3C-score lower than −0.05 (examples in Fig. 1b). The 3C-high class, which is characterised by the highest complementarity, included only a minority of all the predicted interactions ($n = 17988$, 0.8%; Fig. 1c and Supplementary Fig. 1). In addition, the likelihood of interaction between a miRNA and a given target (and the effects of such an interaction) also depends on their relative expression levels[14,15]. Hence, the repertoire of 3C interactions that could affect miRNA stability is strictly context-dependent. We focused on mouse 3T9 fibroblasts, a well-known cellular system particularly apt to the study of transcriptional changes associated with the cell cycle. Indeed, fibroblasts can be induced to quiescence by serum deprivation and then stimulated to re-enter cell cycle by serum addition[30] (Fig. 1d). This fibroblast-serum model offers two key advantages to the study of target-directed miRNA degradation: (i) as changes in gene expression associated with cell cycle re-entry occur over a few hours, miRNAs dilution by cell division does not come into play (ii) data of time course analysis of gene expression and miRNA expression, transcription and decay are already available at high-resolution and at genome wide level[27]. When we scanned our list of target:miRNA 3C-pairs for the ones that could be studied in our model system (e.g., miRNA and target concomitantly expressed in fibroblasts), 18,856 pairs were found, with 1083 falling in the 3C-high class (Fig. 1d). We, then, stratified these pairs according to two parameters influencing the probability of a single target to promote miRNA degradation: (a) the maximum (max) change in expression of each 3C-target observed during cell cycle re-entry (Fig. 1e, $y$-axis); and (b) the max contribution (over the entire time course) of each single target to the total pool of 3C-targets of a given miRNA (% of 3C_Pool; Fig. 1e, $x$-axis). A few 3C-targets stood out in the analysis (Supplementary Fig. 2a), with Serpine1 showing the highest dynamic range (max log₂ fold-change [FC] = 6.92), a very high 3C-score for miR-30b-5p and -30c-5p, and a significant contribution to the 3C-target pool (of 33.8% and 35%, respectively).

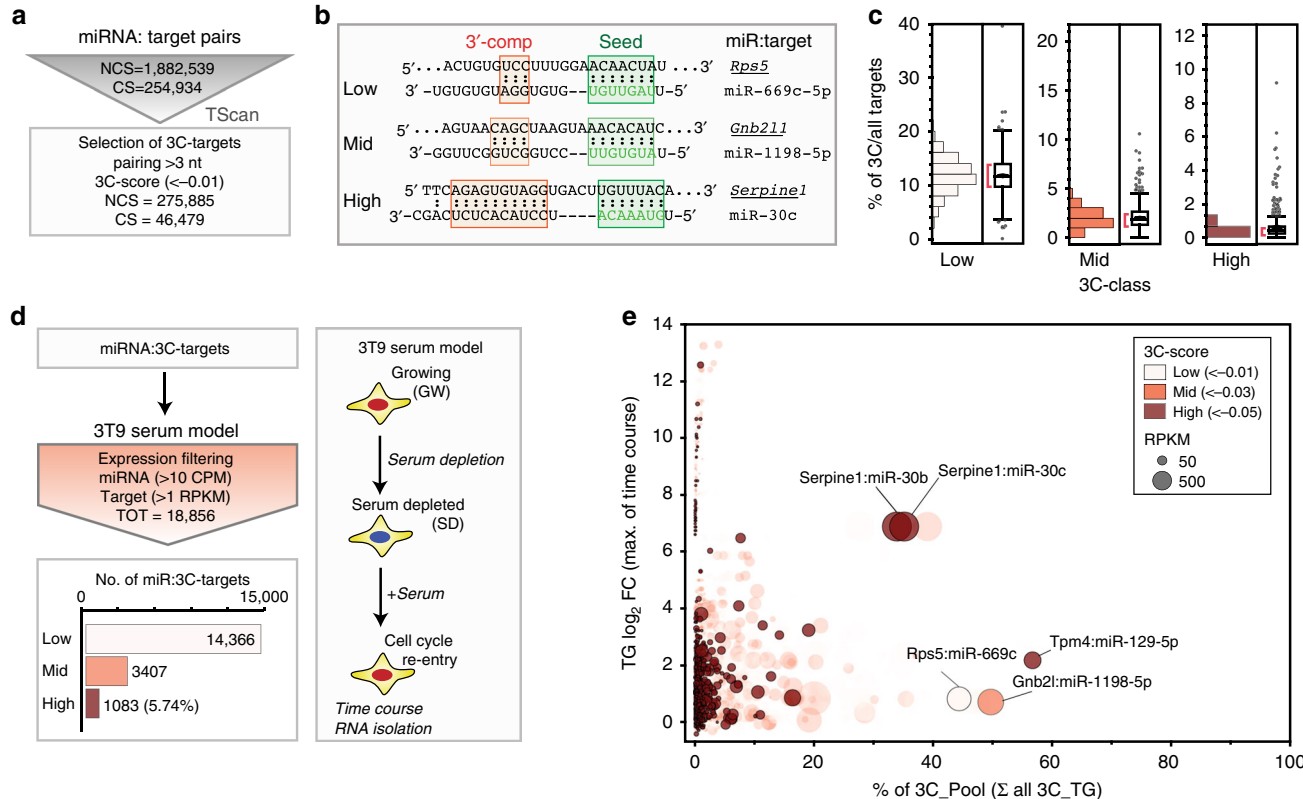

**Fig. 1** Candidate TDMD targets in 3T9 fibroblasts. **a** Scheme summarising filtering criteria to select 3C-targets from all miRNA:target pairs present in the TargetScan6.2 database (NCS non-conserved sequence, CS conserved sequence). All 3C-targets were further classified according to their 3′ complementarity score (3C-score). **b** Three classes of 3C-scores were identified: low, from −0.01 to −0.03; mid, from −0.03 to −0.05; and high, lower than −0.05. Representative examples are shown. **c** The fraction of 3C-targets over the total number of predicted targets, has been calculated for each miRNA and plotted as distribution, with box-plots and whiskers for each class of targets. **d** 3C-targets (as in (**a**)) were screened to determine which ones were eligible for the 3T9 serum model (schematised on the right). Only the pairs expressed in at least one condition (growing cells, serum depleted cells or time course following serum stimulation) were retained. The distribution of filtered pairs based on the 3C-score is also reported. **e** Bubble plot showing all the filtered miRNA:target pairs (from (**d**)) stratified according to the target max contribution (percentage; x-axis) to the 3C-targets pool of a miRNA ($RPKM_{target}/RPKM_{Sum\_all\_3C\_targets}$) and to the target max regulation (log$_2$ FC; y-axis) across the entire time course. Bubble size is proportional to target abundance ($RPKM_{target}$), and target colour reflects the class of 3C-score

**Fast downregulation of miR-30b/c during cell cycle re-entry.** *Serpine1*, also known as *PAI-1*, encodes a serine protease inhibitor protein that inhibits fibrinolysis as well as playing a prominent role in the early G0 > G1 transcriptional programme[30]. All five members of the miR-30 family can interact with Serpine1 transcript, as it has a high-affinity 8-mer seed-match (Fig. 2a). However, it forms an internal bulge (5 nt) and extended 3′ base pairings (9 nt and 11 nt, respectively) only with miR-30b-5p and miR-30c-5p (herein referred to as miR-30b/c) but not with the remaining members of the miR-30 family (miR-30a-5p, miR-30d-5p, and miR-30e-5p, henceforth referred to as miR-30a/d/e). Hence, Serpine1 is likely to act as a decay target only for miR-30b/c. Closer inspection of the time course experiment revealed that Serpine1 expression was potently and quickly induced by serum stimulation of quiescent cells, reaching a peak at 2 h (>1000 RPKM; Fig. 2b, left panel). Serpine1 contribution to the total target pool reached its maximum at this stage, with this single transcript representing almost the entire pool of 3C-high targets for miR-30b/c (>90%; Fig. 2b, right panel and Supplementary Fig. 2a). A concomitant analysis by small RNA sequencing (sRNA-seq) over the time course of stimulation, revealed that miR-30b/c levels rapidly decreased at 4 h after serum addition (log$_2$ FC of −0.88 and −1.13 fold-change, respectively; Fig. 2c), while those of miR-30a/d/e remained almost unchanged.

Target-directed degradation has also been associated with the generation of non-templated modifications at the 3′ end of miR-NAs (tailing and trimming isomiRs[18,19,31]). Using the IsomiRage tool to map isomiRs[32], we found that 3′ tailing of both miR-30b/c had substantially increased, mirroring their downregulation patterns (Fig. 2d), while their trimming variants had been only minimally affected (Fig. 2e). Adenylation and uridylation were the prevalent modifications (Fig. 2f), with about 40% of them being A-forms. The abundance of miR-30a/d/e tailed and trimmed isoforms remained substantially unchanged over the entire time course (Fig. 2d, e). We looked at the transcription of the entire miR-30 family by measuring the synthesis rate of primary transcripts during serum stimulation through a pulse labelling approach[33]. It should be noted that miR-30b and -30c (which has two copies, 30c-1 and 30c-2) belong to two different genetic units and, therefore, are not co-transcribed (Fig. 2g). During our time course experiment, pri-miR-30b-30d and pri-miR-30e-30c-1 synthesis rates did not change considerably, while pri-miR-30a-30c-2 transcription initially decreased and then increased slowly (Fig. 2g). No alterations were observed when looking at passenger miRNAs (Supplementary Fig. 2b). Taken together, these data suggest that the specific and quick reduction in miR-30b/c abundance during serum-induced cycle re-entry is likely the result of post-transcriptional events involving miRNA degradation.

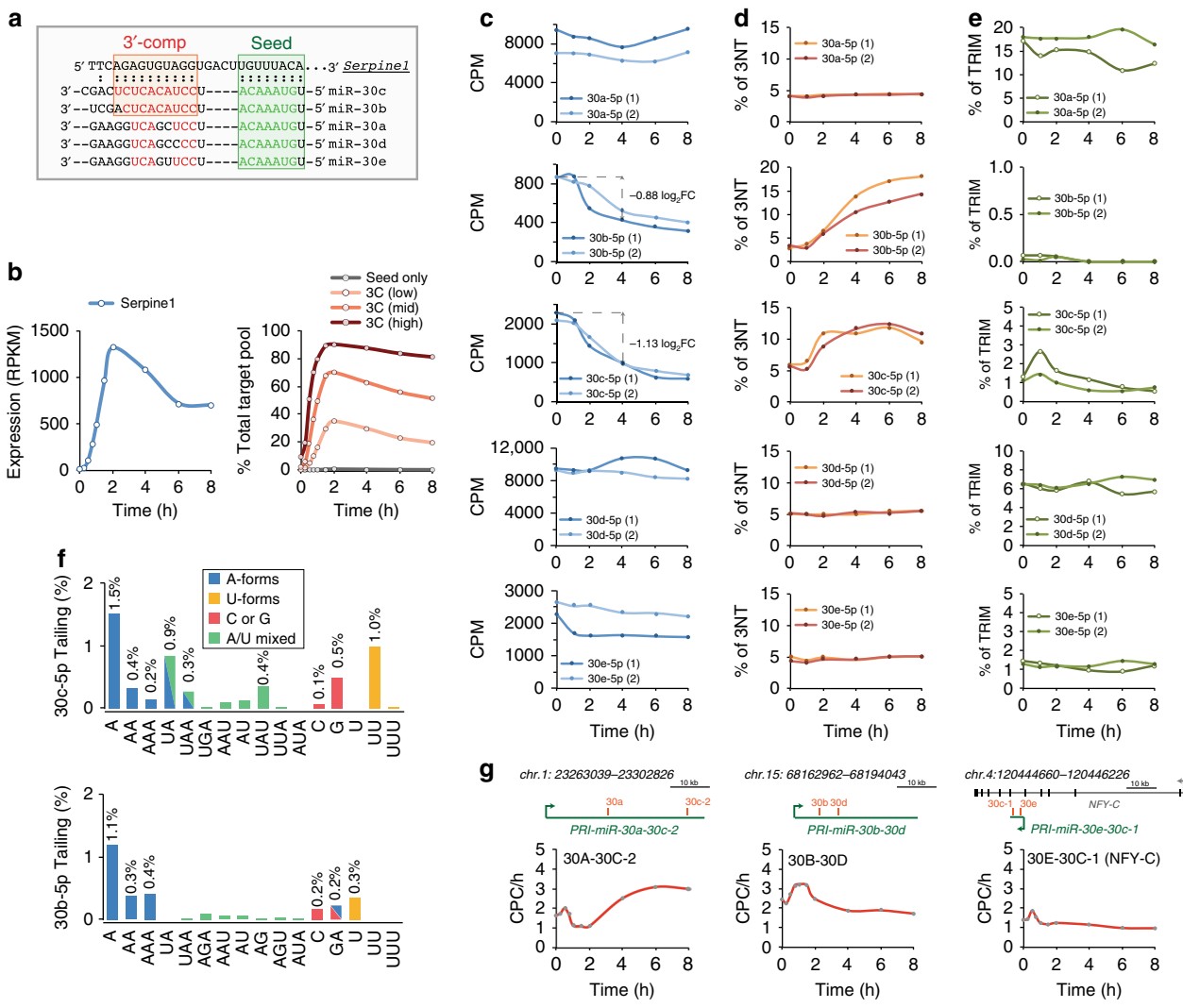

**Fig. 2** Regulation of Serpine1 and miR-30 family members during serum stimulation. **a** Alignment of Serpine1-MRE with miR-30 family members. The seed-match (identical for all miR-30 family members) is shown in green, and the bases of 3′ complementarity, which is extensive only for two miRNAs (miR-30b, 9 nt; and miR-30c, 11 nt), in red. **b** Expression of Serpine1 upon serum stimulation of quiescent 3T9 fibroblasts. Left panel, RPKM. Right panel, contribution of Serpine1 expression to the total target pool, calculated over different classes of 3C-score. **c–e** Expression of miR-30 family members upon serum stimulation of quiescent 3T9 fibroblasts. Shown for each miRNA are: **c** the expression level (counts per million, CPM), **d** tailing events (% of 3′-non-templated isomiRs, 3NT) and **e** trimming events (% of trimmed forms, TRIM) in two independent experiments. **f** Frequency of tailing (3′NT isomiRs) for miR-30c-5p (top panel) or miR-30b-5p (bottom panel) in asynchronously growing 3T9 cells. Percentages of total reads. **g** Synthesis rate of the primary transcripts of the miR-30 family upon serum stimulation, as measured by a short pulse of 4SU. Definition of genomic regions and calculation of synthesis rate as copies per cell per hour (CPC/h) have been described previously[27]

**Serpine1 controls miR-30b/c abundance in the cell**. The range of Serpine1 and miR-30s expression levels in the cell was rigorously investigated by absolute quantification (copies per cell, cpc). To this aim, a titration curve was obtained using synthetic miR-30 RNA oligonucleotides and a Serpine1 plasmid DNA template (Supplementary Fig. 3a–c). In exponentially growing cells (GW) miR-30c was expressed at ~70 cpc (confirmed also by droplet-digital PCR, Supplementary Fig. 3d) and miR-30b/c accounted for nearly 40% of the total miR-30 copies (Fig. 3a). In growing cells and quiescent fibroblasts, miR-30c and Serpine1 were expressed at comparable levels (Serpine1: 170 cpc in GW cells and 88 cpc in quiescent cells; Fig. 3b), with a target-per miRNA (TPM) value of ~1 (Fig. 3c). Upon serum stimulation, Serpine1 increased enormously, peaking at > 4000 cpc (at 4 h) and a >40 TPM ratio (Fig. 3c). Hence, Serpine1 met both molecular and quantitative requirements for eliciting miR-30b/c degradation. In order to directly verify the occurrence of TDMD, we devised a strategy

using the CRISPR/Cas9 system[34] to manipulate Serpine1-MRE. We designed sgRNAs that would allow the deletion of a minimal region at the Serpine1 3′UTR (~160 nt, Fig. 3d), inclusive of miR-30 MRE. Poorly conserved MREs for miR-224 and miR-320 were also part of the deleted region, but their corresponding miRNAs are negligibly expressed in 3T9 cells. We generated two independent mutant clones (Serpine1:miR-30[MRE-KO1] and Serpine1:miR-30[MRE-KO2], hereafter referred as MRE-KO1 and MRE-KO2) (Supplementary Fig. 4a, b). In neither clones, Serpine1 mRNA and protein expression resulted affected by the manipulation of the gene 3′UTR (Fig. 3e, f). In complete medium, both clones showed proliferation rates and cell cycle profiles similar to those of their parental cells (Supplementary Fig. 4c, d). When measuring global expression of miRNAs and their isoforms by sRNA-seq, we noticed that, in both MRE-KO clones, only a few miRNAs were significantly induced, including miR-30c-5p (1.03 log₂ FC, $p = 0.0013$) and 30b-5p (0.92 log₂ FC, $p = 0.0180$) (Fig. 3g,

Supplementary Fig. 4e, Supplementary Data 2). Remarkably, expression levels remained unchanged when looking either at members of the miR-30 family lacking the extended 3′ complementarity, or at passenger miRNAs. Quantitative reverse transcription PCR (RT-qPCR) was performed to measure primary transcripts levels for the three miR-30 loci and no difference was found between parental cells and MRE-KO clones (Fig. 3h),

suggesting that the observed increase in miR-30b/c levels was due to a decrease in their degradation. Consistently, tailing—one of the hallmarks of target-directed degradation—of miR-30b/c decreased in MRE-KO clones, with adenylation being the most affected modification (Fig. 3i). To rule out the possibility that increased miR-30b/c was a by-product of clonal selection, we used RT-qPCR to measure miR-30c absolute levels in clones with

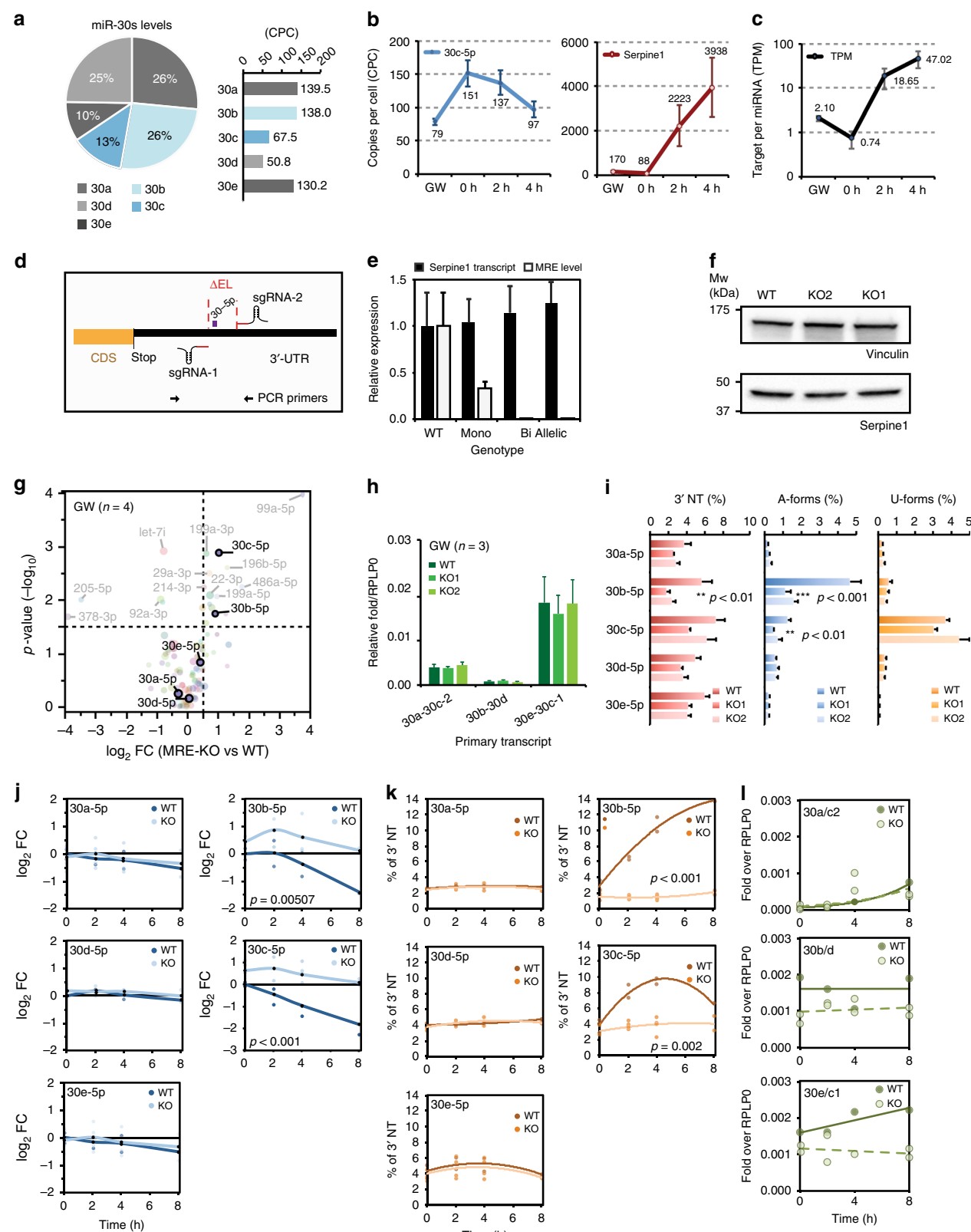

no MRE deletion (no-del clones) and found them to be the same as in their parental cells (~80 cpc; Supplementary Fig. 4f). In MRE-KO clones, miR-30c absolute levels instead doubled (184.7 cpc; Supplementary Fig. 4f). We also investigated miR-30 levels over the time course of serum-stimulation in MRE-KO cells. As expected, in parental cells, miR-30b/c levels were quickly downregulated upon serum stimulation (Fig. 3j), with a rapid increase of 3′ tailing forms (4–8 h post stimulation) (Fig. 3k; see also Fig. 2d). In MRE-KO cells, miR-30b/c levels instead remained consistently high and no sudden increase in tailing variants was observed (Fig. 3k). While the abundance of U-forms did not change, the induction of A-forms (the large majority of tailing isoforms in parental cells) was totally abolished in MRE-KO cells (Supplementary Fig. 5b). The levels and modifications of other members of the family (i.e., 30a-d-e) were unchanged, and primary transcripts from all the three miR-30 loci stayed at similar levels over the time course (Fig. 3l), further confirming a post-transcriptional regulation of miR-30b/c by Serpine1-MRE. In conclusion, under both steady-state (growing cells) and dynamic (serum stimulation) conditions, miR-30b/c levels are controlled by Serpine1 via a target-directed degradation mechanism.

**Wild-type but not mutant MRE rescues TDMD**. We next sought to rescue the interaction between miR-30 and Serpine1 3′ UTR in MRE-KO cells. To this aim, we used an adenoviral construct expressing a red fluorescent tracking protein (RFP) fused to Serpine1 3′UTR (Fig. 4a). We used Serpine1 wild-type sequence (SE1-WT), with its miR-30 MRE, or mutant sequences, in which the miR-30 MRE was completely replaced (SE1-MUT) or only mutated at the level of the 3′ complementary region, leaving the match with the miRNA seed (SE1-SEED) (Fig. 4a). First, we analysed the effects of the three constructs by RT-qPCR. The RFP-SE1-WT construct significantly reduced the levels of miR-30c and, to a lesser extent, miR-30b, without affecting any other member of miR-30 family or the unrelated miRNA let-7b (Fig. 4b). In particular, miR-30c levels decreased moderately (~20%) when RFP-SE1-WT construct was expressed at moderate levels (~1000 cpc, with a transduction efficiency of ~30% RFP + cells) and considerably (~3-fold) when the same construct was expressed at high levels (~10,000 cpc; nearly 100% RFP + cells) (Fig. 4c, d). No effects were elicited by any concentration of either of the two mutant constructs (SE1-MUT and SE1-SEED), confirming that the 3′ complementarity is indispensable for target-directed degradation (Fig. 4c, d). These results were fully reproduced when miRNA expression was analysed at a global level by sRNA-seq (Fig. 4e). Expression of the RFP-SE1-WT construct affected also the level of tailing variants, only in the case of miR-30c and miR-30b, with a marked preference for adenylation

(Fig. 4f and Supplementary Fig. 5b). Using a set of Serpine1 mutants with various levels of 3′ complementarity to miR-30c (Fig. 4g), we investigated 3′ pairing rules in the context of ectopic MRE expression. As previously observed[18], four mismatches in the 3′ end were sufficient to nearly abolish TDMD (M4-M6 mutants, Fig. 4h), while the central bulge could be extended to up to seven nucleotides with only a modest reduction in TDMD (B7 mutant, Fig. 4h).

**miR-30 activity is modulated by Serpine1**. Endogenous targets involved in TDMD are also supposed to modulate the global activity of miRNAs by inducing de-repression of shared targets (Fig. 5a), similarly to ceRNAs[13,15]. Conversely, TDMD suppression (as in MRE-KO cells) should result in increased miRNA activity. Using a highly sensitive reporter system to measure miR-30c activity at single-cell level ('miR-sensor', Supplementary Fig. 6a) we observed that repression of the synthetic target increased from 5.5-fold in parental cells to 7.7-fold in MRE-KO cells ($p < 0.001$; Supplementary Fig. 6b, c). Higher levels of miR-30c had thus resulted in a significant increase in its activity. To gain a deeper understanding of miRNA activity at a global level and on endogenous targets, we measured the full transcriptome of parental and MRE-KO cells, in steady-state conditions (growing–GW) as well as following serum stimulation (TC) (GSE104648; Supplementary Data 3). Looking at genes that were expressed in both conditions (GW and TC, $n = 9619$), we isolated miR-30 targets based on: (i) predictions by computational approach (TargetScan7.1: 899 conserved, CS, and 1795 non-conserved, NCS[29]); (ii) experimentally supported interactions ($n = 1037$ for miR-30 s) obtained by High-throughput Sequencing of RNA isolated by crosslinking immunoprecipitation (HITS-CLIP-from TarBase7.0[35]); and (iii) miRNA:target chimaeras ($n = 1037$ for miR-30c-5p), obtained by sequencing of Covalent Ligation of Endogenous Argonaute-bound RNAs (CLEAR-CLIP[36], Supplementary Data 3). First, we evaluated miR-30 targets dynamics in wild-type cells. As miR-30b/c levels became strongly down-regulated following serum addition (see Fig. 2c), coherently, miR-30 targets were more induced (compared to background genes) at 8 and 12 h after serum stimulation ($p < 0.0001$, Supplementary Fig. 7a–c). We, then, compared target repression in MRE-KO and wild-type cells. In growing conditions, miR-30 targets resulted slightly ($-0.05 \log_2$ FC) but significantly more repressed in mutant cells (Fig. 5b, d, f). The repression was generally stronger for high-affinity MREs (8-mers and 7-mers-m8) than for low-affinity MREs (7-mers-A1) (Fig. 5b, c). Following serum stimulation, in MRE-KO cells, the downregulation of miR-30b/c has been shown to be abrogated (see Fig. 3j). In this dynamic setting, the repression of miR-30 targets became stronger ($p < 0.0001$, Fig. 5e, g, i),

**Fig. 3** Serpine1:miR-30b/c interaction controls miR-30b/c degradation. **a** Percentage contribution and copies-per-cell for each member of the miR-30 family in growing cells are shown, based on absolute quantification data in Supplementary Fig. 3a–c. **b** Absolute quantification of miR-30c-5p and Serpine1 in 3T9 model. Data refer to three independent biological experiments (average and s.e.m). **c** A target-per-miRNA (TPM) value was calculated based on absolute quantification. **d** Strategy for the CRISPR-Cas9-mediated removal of miR-30 MRE from Serpine1 3′UTR. Shown are the positions of the sgRNAs and the PCR primers used for clone sequencing (see also Supplementary Fig. 4a, b). **e** Expression levels of Serpine1 by RT-qPCR in wild-type (WT) 3T9 fibroblasts and in clones with mono-allelic ($n = 3$) or bi-allelic deletion of the MRE for miR-30 (MRE-KO1 and MRE-KO2; shown individually). Different primers pairs were used to quantify total Serpine1 transcripts or just those with an intact MRE (see Methods). **f** Serpine1 protein level in WT or bi-allelic MRE-KO clones. Vinculin was used to normalise sample loading. **g** sRNA-seq was performed to measure miRNA expression in WT ($n = 4$) and bi-allelic MRE-KO clones (KO1 and KO2; $n = 3$ each) in asynchronously growing conditions. The graph shows the average $\log_2$ FC (MRE-KO clones as compared to WT cells; x-axis) and the p-value ($-\log_{10}$; y-axis), by Welch's t-test. Colour coding reflects different miRNA families. **h** Primary transcripts for the miR-30 family measured by RT-qPCR in WT or MRE-KO clones ($n = 3$). **i** Total frequency (%) of tailing events (3′NT), or adenylated or uridylated only isomiRs (A- and U-forms), measured by sRNA-seq in WT ($n = 4$) or MRE-KO clones ($n = 3$); in **h**, **i** mean and s.d., asterisks mark significant values (**$p < 0.05$, ***$p < 0.001$; Welch's t-test). **j**, **k** Expression of miR-30 family by sRNA-seq upon serum stimulation in WT and MRE-KO cells. For each miRNA, expression regulation ($\log_2$ FC; **j**) and tailing events (% 3′NT; **k**) are shown. Details of miRNA isoforms in Supplementary Fig. 5. **l** Primary transcripts of the miR-30 family by RT-qPCR. **j**, **k**, **l** two independent experiments shown

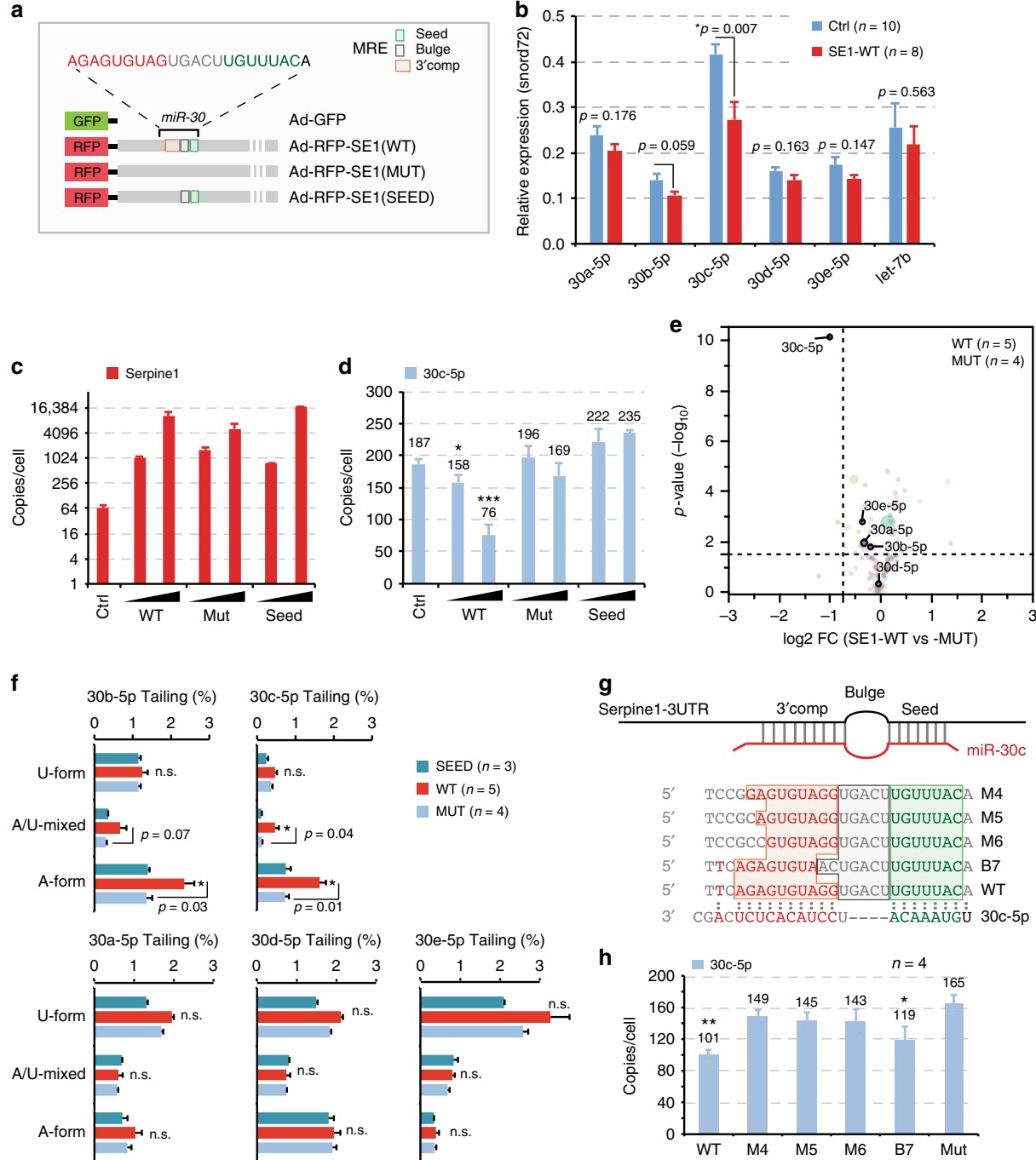

**Fig. 4** Re-expression of Serpine1-MRE rescues miR-30c degradation. **a** Scheme of the recombinant adenoviral constructs used to re-express Serpine1 3′ UTR fused to an RFP reporter (rescue experiments in (**b–f**)), showing the wild type form of the 3′UTR of Serpine1 (WT) and two mutants, one where miR-30 MRE had been completely abrogated (MUT) and the other where the 3′ supplementary site had been mutated but the seed sequence left intact (SEED). **b** Bar plot showing the expression of miR-30 family (and Let-7b as a control) by RT-qPCR in cells infected with control virus (Ctrl) or with Ad-RFP-SE1 (WT) adenoviruses. *p*-values were determined by Welch's *t*-test. **c**, **d** Absolute expression (copies per cells) of Serpine1 3′UTR transcripts and miR-30c-5p measured by RT-qPCR in the rescue experiments. Adenoviruses were expressed at two different multiplicity of infection (MOI ~1 and ~5). Asterisks mark significant values (*$p < 0.05$, ***$p < 0.001$; Welch's *t*-test) **e** sRNA-seq was performed to measure miRNA expression in MRE-KO cells that express a wild type (WT, $n = 5$) or a mutated (MUT, $n = 4$) 3′UTR of Serpine1. The graph shows the average $\log_2$ FC (*x*-axis) and the *p*-value ($-\log_{10}$, *y*-axis) by Welch's *t*-test. Colour coding reflects different miRNA families. **f** Frequency (%) of tailing events, distinguished as adenylated (A-), uridylated (U-), or mixed isomiRs (A/U-mixed), as measured by sRNA-seq. *p*-values by Welch's *t*-test. **g** Scheme of the mutant Serpine1 constructs (see also Methods) that were used to dissect TDMD 3′ pairing rules (**h**). Seed in green and 3′ supplementary pairing in orange. **h** Absolute expression (copies per cells) of miR-30c-5p measured by RT-qPCR upon over-expression of different Serpine1 mutants (shown in (**g**)). Cells were collected at 36 h after transfection (*$p < 0.05$, ***$p < 0.01$, Welch's *t*-test)

with the median repression reaching –0.2 log$_2$ FC at both 8 and 12 h post stimulation (Fig. 5b, c). The specificity of the effect was confirmed by comparing the median regulation of the targets for all expressed miRNAs (Fig. 5j, k and Supplementary Fig. 7d): miR-30 targets displayed preferential repression during serum stimulation but not in growing cells. A certain degree of specificity

emerged for targets of miR-30b and -30c, as defined according to CLEAR-CLIP miRNA:target chimaeras (Fig. 5k), suggesting that miR-30b/c preferentially binds to targets with a supplementary pairing in the 3′ region, as previously observed[36,37].

Taken together, these data strongly support a role for the Serpine1:miR-30b/c interaction in modulating the activity of

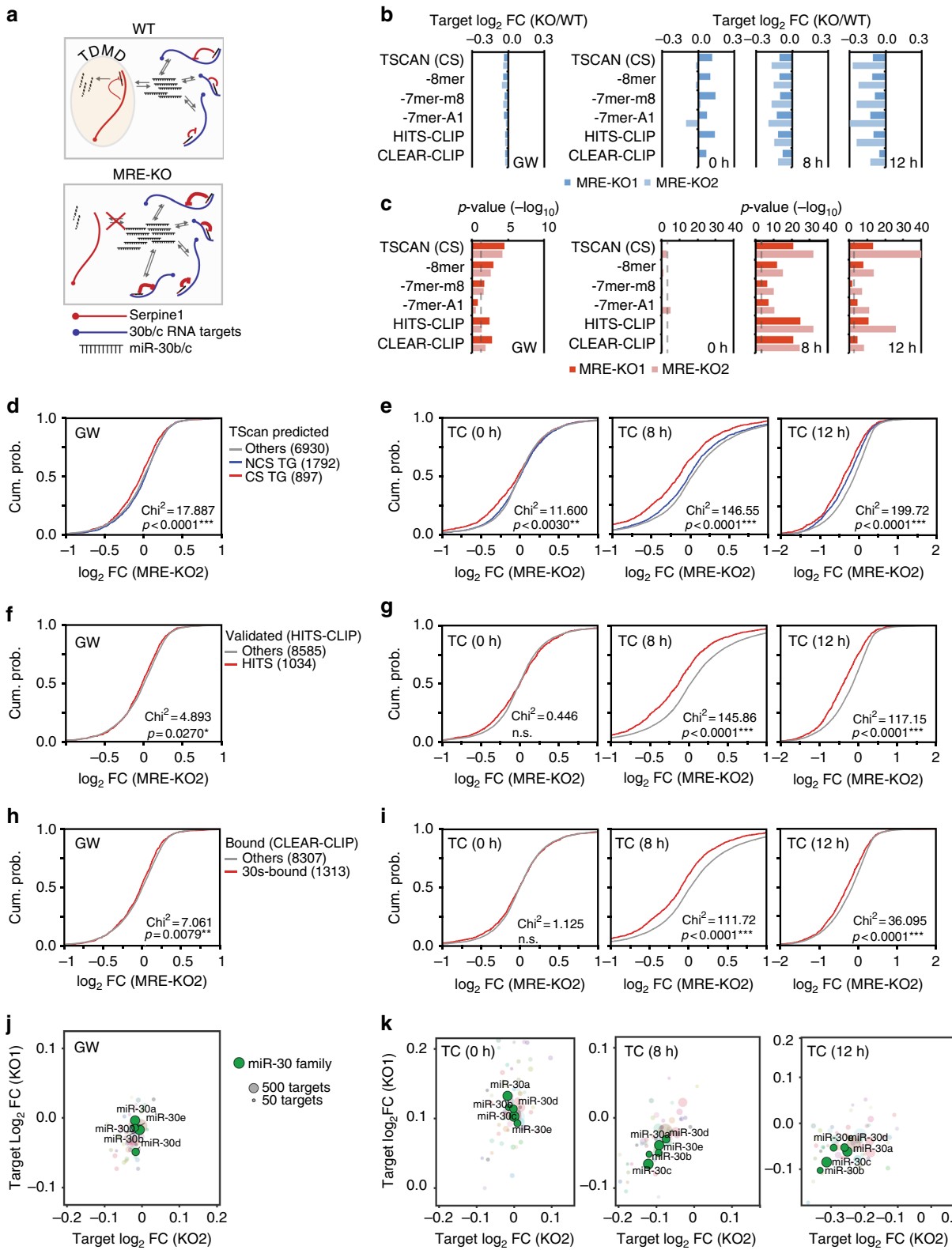

miR-30 on shared targets, particularly by temporally restricting miRNA activity during cell cycle re-entry of quiescent fibroblasts.

**Serpine1:miR-30b/c interaction affects cell behaviour.** We next investigated whether the increased miR-30 activity observed when the Serpine1:miR-30b/c interaction is lost, might also cause phenotypical alterations (Fig. 6a). Previous works have linked the miR-30 family to apoptosis sensitivity and p53 signalling[38,39]. Thus, we speculated that MRE-KO cells might be sensitised to cell death. While the basal level of apoptosis did not seem to change in growing MRE-KO cells (Fig. 6b, d), it appeared largely increased when we exposed these mutant cells to stress conditions, such as serum starvation or treatment with the anthracycline drug doxorubicin (Fig. 6c, d). Analysis of cell cycle re-entry dynamics had previously indicated that the expression levels of miR-30 targets are significantly affected in MRE-KO clones (see Fig. 5). BrdU incorporation experiments now revealed that these mutants show a remarkable acceleration in S-phase entry, (Fig. 6e, f), suggesting that G1/S transition becomes faster in the absence of a Serpine1-directed miR-30b/c degradation mechanism. When performing time-lapse microscopy analysis, we saw that MRE-KO cells had an accelerated mitotic rate (Fig. 6g and Supplementary Movies 1–3), confirming that these mutants re-enter cell cycle more quickly than the wild type. Notably, a significant number of MRE-KO cells also underwent apoptosis (Fig. 6h). Through gene expression analysis aimed at identifying differentially expressed genes (DEGs) in MRE-KO cells, we identified 244 DEGs in growing cells (steady-state) and 1648 DEGs in quiescent cells that had undergone serum stimulation (Fig. 7a, Supplementary Data 3 and Methods). At steady-state, we found an enrichment of genes belonging to the p53 pathway and the apoptotic response (Fig. 7b), which might explain the increased sensitivity to cell death of MRE-KO cells upon stress induction. No enrichment of miR-30 targets among these 244 DEGs, suggesting that the observed cellular behaviour is also due to secondary effects that originate from the disruption of the Serpine1-mediated degradation of miR-30b/c. As for the 1648 DEGs identified in stimulated cells, we used unsupervised clustering to classify them according to their temporal pattern of regulation (cluster I–V; Fig. 7c, left panel) and, based on the dynamics of serum stimulation, to distinguish constitutive (cluster I and V) from acute effects (cluster II, III, IV) (Fig. 7c, right panels). We then associated each cluster with the correspondingly enriched biological properties (Fig. 7d). In particular, compared to wt cells, MRE-KO cells showed earlier induction (cluster II) of several E2F-regulated genes involved in cell cycle and mitosis. This might explain a quicker entry into S-phase and mitosis of mutant cells (see Fig. 6e–g). Notably, clusters IIb (serum-induced genes with an earlier induction in mutant cells;

e.g., Ccne2) and IV [genes repressed in mutant cells at late (8h-10h-12h) time-points] were significantly enriched with miR-30 direct targets (Fig. 7e, f). In particular, cluster IV comprises miR-30 targets that are key regulators of p53-pathway (Bcl6, Sirt1) and of the circadian clock (Per2, Cry2), which in turn is coupled with the cell cycle[40]. In summary, the Serpine1:miR-30b/c interaction profoundly affects cell behaviour by increasing sensitivity to apoptosis and by accelerating the G1/S transition during cell cycle re-entry of quiescent cells.

**Discussion**

Here, we provide molecular evidence that in mammalian cells a target-directed miRNA degradation mechanism exists and can control miRNA activity and cell functions. We found hundreds of targets that are potentially capable of TDMD. Focusing our study on Serpine1, we provide a proof-of-principle characterisation of TDMD at endogenous levels of expression, as opposed to acute expression of artificial targets or ectopic over-expression. Indeed, we show that under physiological conditions (i.e., both in growing fibroblasts and in quiescent cells undergoing serum stimulation) Serpine1 post-transcriptionally regulates two closely related miRNAs, miR-30b, and miR-30c, by inducing their degradation. Degradation promoted by Serpine1 is characterised by some peculiar features. It involves specific post-transcriptional modifications at the miRNA 3′ end, with higher-than-normal levels of adenylation but no uridylation involved, as previously reported[18]. Finally, it only affects two members (miR-30b and 30c) of the conserved miR-30 family that are not co-transcribed, thus coupling them post-transcriptionally while uncoupling them from the other three miR-30 members (miR-30a/d/e). All miR-30s can interact with Serpine1 via their seed sequence (which is shared by all miR-NAs in the same family), but only miR-30b/c present an extended 3′ end complementarity (3C pairing) to Serpine1 transcript. Therefore, as previously observed in vitro and with artificial targets[18,20], 3C pairing is essential in the case of a TDMD mechanism triggered by endogenous targets. Accordingly, Serpine1 has stronger effects on miR-30c than on miR-30b, as miR-30c forms additional 3′ base pairings with Serpine1. The importance of 3C pairing is highlighted also by the fact that it is required in reconstitution experiments. Additional mismatches can be tolerated close to the bulge, but completely impair miRNA degradation if located in proximity of miRNA 3′ end. Complementarity is not the only requisite for TDMD, which also presents some context-dependent quantitative requirements. Therefore, we performed absolute measurements of miRNA and target expression (copies per cells) and computed the corresponding target-per miRNA value (TPM) to investigate the stoichiometry of their interaction. Upon serum stimulation of quiescent cells, Serpine1 is strongly induced so to

**Fig. 5** miR-30 activity increases in Serpine1-MRE-KO cells. **a** Scheme representing WT and MRE-KO cells expected behaviour in terms of miR-30c activity with canonical targets (in blue) and decay targets (in red). When TDMD is suppressed (i.e., by removing the MRE from a TDMD target, like Serpine1), miRNA increases in abundance and is redistributed on targets thus increasing its repression activity. **b**, **c** Expression analysis of WT and MRE-KO cells was performed by RNA-seq in different conditions of cell growth (GW: growing cells; 0 h: quiescent cells; 8 h and 12 h: serum-stimulated cells). **b** Bar charts report the difference between the median log2 fold changes (MRE vs WT) of different subsets of miR-30 targets. The analysis was performed with targets either predicted by TargetScan7.1 (897 conserved targets, CS, also broken down by seed-match as 8-mer, 7-mer-m8, or 7-mer-A1) or validated by HITS-CLIP (1037 targets, from Tarbase v7.0[35]) and CLEAR-CLIP (1315 targets[36]). **c** p-values of data shown in (**b**). **d**, **e** Cumulative distribution functions (CDF) of $\log_2$ FC of mRNA (MRE vs WT) for conserved targets (CS), not conserved targets (NCS) and genes not predicted as targets (Other). **f**, **g** CDF plot as in (**d**, **e**) with miR-30 targets from HITS-CLIP experiments[35]. **h**, **i** CDF plot as in (**d**, **e**) with miR-30 targets from CLEAR-CLIP experiments[36]. **j**, **k** Targets for all expressed miRNAs were identified by CLEAR-CLIP. The bubble plots report the median log2 fold changes (MRE-KO vs WT), calculated for every miRNA having at least 50 expressed targets. Colour code is for miRNA families. Members of the miR-30 family are highlighted in green. All statistical analyses (chi-square and p-value) were performed using the Wilcoxon test

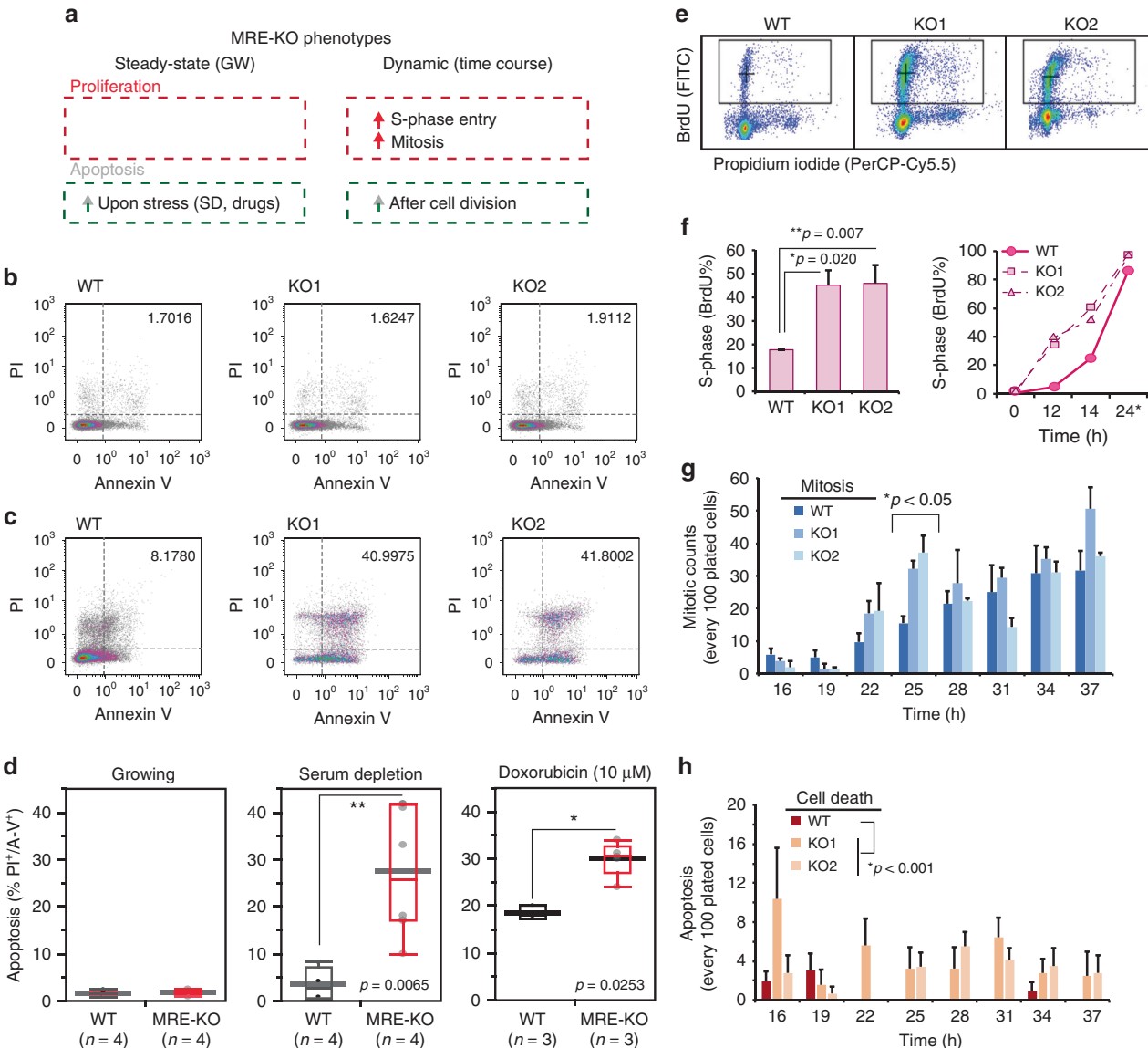

**Fig. 6** MRE-KO cells display an altered cell cycle re-entry and are sensitised to apoptosis. **a** Summary of the phenotypes of MRE-KO cells. **b**, **c** Sensitivity to apoptosis was analysed in MRE-KO clones and WT cells by FACS profiling; apoptotic cells were identified by propidium iodide (PI) and Annexin-V staining; **b** asynchronously growing condition; **c** cells were serum depleted for 48 h. **d** Box-plots of apoptotic cells (PI$^+$ and Annexin-V$^+$) in WT and MRE-KO clones in different conditions. Asterisks mark significant values (*$p < 0.05$, **$p < 0.01$, Wilcoxon test). **e** Dynamics of cell cycle re-entry induced by serum stimulation by FACS analysis. S-phase entry was determined by BrdU incorporation (30 min pulse) and propidium iodide staining. **f** Left: S-phase entry evaluated at 14 h post–serum stimulation (average and s.e.m.; $n = 3$). Right: line chart shows a single time course of cell cycle re-entry (24*: cells treated in continuous with BrdU for 12 h). **g**, **h** Number of mitosis (**g**) and cell deaths (**h**), normalised for every 100 plated cells, occurring between 16 and 37 h after serum stimulation as analysed by time-lapse microscopy ($n = 3$). **f**, **g**, **h** $p$-value by Welch's test

largely exceed the levels of miR-30b/c (TPM > 10) and very rapidly halve their levels (i.e., half-life of ~4 h). No changes in transcription or processing are concomitantly observed (see Fig. 2). This effect is completely dependent on Serpine1, as miR-30b/c repression is abolished in MRE-KO cells. Thus, a sudden and massive upregulation of a TDMD target can have a role in accelerating miRNA degradation within a precise time window. Indeed, we had previously estimated a longer half-life for miR-30b and -30c (~12 and 16 h, respectively) in growing fibroblasts[27]. In growing cells Serpine1 is expressed at relatively low levels (170 cpc) and in range with. the number of miRNA copies (about 70–80 cpc for miR-30c and 140 for miR-30b). Nevertheless, miR-30b and miR-30c are subjected to TDMD also in growing cells, as in MRE-KO cells the levels of the two

miRNAs almost doubled. Therefore, endogenous TDMD takes place even for targets that are present at relatively low levels of expression, provided that they are in a stoichiometric range with their cognate miRNAs (TPM ≥ 1). The reason being that every time the miRNA interacts with the target, it is irreversibly subtracted from the miRNA pool by degradation[16]. It is worth mentioning that in reconstitution experiments, where targets are provided ectopically, TDMD requirements differ slightly as Serpine1-MRE acts mostly on miR-30c-5p and, in order to do so, must reach higher levels of expression (at least ~1000 cpc). In these experiments, miRNA levels were analysed at 36 h post infection. It is conceivable that a longer time interval is necessary to observe a significant reduction in miR-30b abundance, as this is expressed at relatively higher levels than

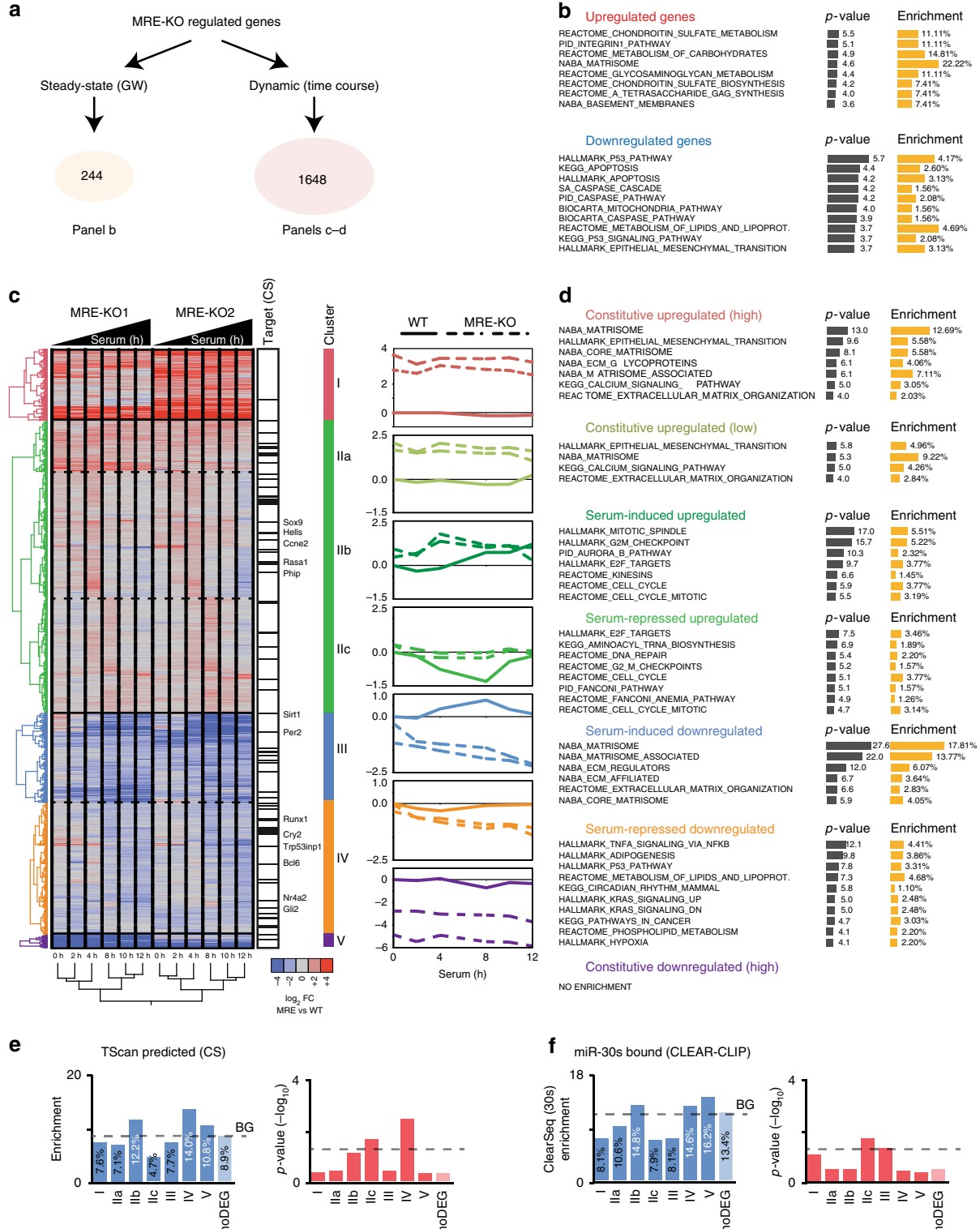

**Fig. 7** MRE-KO cells display altered gene expression. **a** Differentially expressed genes (DEGs) between WT and MRE-KO clones were determined by RNA-seq either at steady-state (growing cells, $n = 244$) or upon serum stimulation ($n = 1648$) of quiescent cells. **b** Enriched biological functions among GW-DEGs were retrieved by Gene Set Enrichment Analysis (GSEA), calculating the overlap with curated gene sets (Molecular Signatures Database; shown those with a $p$-value < 0.001). **c** Unsupervised hierarchical clustering of serum stimulation DEGs (1648 genes) identifies five major clusters (I–V) showing different patterns of gene regulation. Cluster II was further split into three sub-clusters (IIa-b-c) having different behaviour in terms of serum-regulation dynamics. MiR-30 family conserved targets predicted by TargetScan are also reported. Right charts: median gene regulation in each (sub)cluster is reported over the time course; MRE-KO data were normalised over basal expression level (i.e., quiescent state, 0 h) of WT cells. **d** Enriched biological functions among clusters shown in **b**. **e**, **f** Enrichment (and chi-square, $p$-value by Contingency Test) of miR-30 targets across each cluster was calculated either using predictions (**e**) or validated interactions (**f**)

miR-30c. It should be also stressed that, at low MOI, some cells were uninfected and did not ectopically express Serpine1-MRE (as determined by monitoring RFP). This might explain why ectopically expressed Serpine triggered TDMD at higher levels (at least ~1000 cpc) when compared to endogenously expressed Serpine1 (170 cpc).

How many endogenous TDMD targets do exist? We chose to study Serpine1 as it was one of the top hits in a list of candidate targets displaying ideal structural characteristics: a high-affinity seed (8-mer or 7-mer-m8), a central bulge ( > 2 and < 5 nts) and a high 3C-score (high degree of complementarity at the 3′ end). Based on our preliminary observations, we can anticipate that at least 1000 target:miRNA pairs (e.g., 0.05% of all predicted targets) present molecular characteristics that make them suitable for TDMD. While this work was under revision, a different case of endogenous TDMD was provided by Bitetti and co-workers[17]. In their study, the NREP transcript promotes degradation of miR-29b in neuronal cells. The NREP:miR-29b and Serpine1:miR-30b/c pairs share the same complementarity features. Significantly, NREP is almost exclusively expressed in the brain (where TDMD is very effective) and imparts a spatial restriction to miR-29b expression in the cerebellum. Loss of the NREP:miR-29b interaction generates an in vivo phenotype characterised by impaired motor functions in fish and mice. A relevant role for TDMD in animal behaviour has thus been claimed. However, the impact of NREP on shared miR-29 targets was not investigated. Endogenous targets involved in TDMD are expected to influence miRNA-mediated repression. By investigating the Serpine1:miR-30 case, we provide compelling evidence in favour of such hypothesis. MRE-KO cells show a significant increase in miR-30 activity, as confirmed using both a synthetic reporter directly monitoring miRNA activity (miR-sensor) and endogenous targets identified by prediction algorithms or experimental approaches. In growing conditions (at steady-state), effects on endogenous targets are mild (with a ~5% increase in repression) and no major changes in phenotype are observed. Gene expression alterations observed in this context are likely to be a consequence of clone adaptation to the new (increased) levels of miR-30b/c. On the contrary, MRE-KO cells show remarkable phenotypes when exposed to acute stresses. In particular, upon serum stimulation mutant cells fail to properly regulate miR-30b/c levels, enter more quickly into S-phase and display an increased mitotic and cell death rate. In this context, strong (~20%) and specific repression of miR-30 targets is observed, with a certain degree of specificity towards miR-30b/c preferential targets (as opposed to shared miR-30 targets). Indeed, two independent studies mapping miRNA:target interactions suggest that miR-30b/c preferentially bind those targets that possess supplementary pairing in the 3′ region[36,37]. In future, it would be worth trying to identify such specific targets and characterise them in serum-stimulated cells (e.g., by CLASH), as they might help clarify miR-30b/c function in cell cycle re-entry. In conclusion, we have provided evidence that in mammalian cells endogenous targets can control miRNA levels post-transcriptionally and thus modulate miRNA activity on all their targets.

We can no longer consider miRNA targets as being all equal. The way has now been opened to the identification of new endogenous TDMD targets that might be involved in different physiological or pathological processes. TDMD confers a further level of flexibility to miRNA regulation. It can restrict miRNA expression to a specific subset of cells (regulation in space—as for NREP:miR-29b in the cerebellum[17]) or impart temporal-restriction to miRNA activity in dynamic settings (regulation in time—as for Serpine1:miR-30b/c during cell cycle re-entry).

## Methods

**Statistical analysis**. Analyses (Oneway, Scatter Plot, Contingency) and statistics were produced using JMP 12 (SAS) software. Microsoft Excel was used to generate bar graphs with average and standard deviation (s.d.) or standard error mean (s.e. m.) of repeated experiments. The number of replicates and the statistical tests used are indicated in the figure legends.

**Cell treatments and procedures**. Mouse 3T9 fibroblasts a kind gift of Bruno Amati. Cells were cultured in DMEM (Dulbecco's modified Eagle's medium) supplemented with 10% fetal bovine serum (FBS), 100 U/ml penicillin and 100 mg/ml streptomycin. Cells were routinely checked for mycoplasma contamination and resulted negative. 3T9 cells were induced to quiescence in a serum-deprived medium (0.1% FBS) for 3 days. Cell cycle re-entry was stimulated by adding fresh medium with 10% FBS. For cell cycle analysis, cells were treated with 33 μM 5-bromo-2′-deoxyuridine (BrdU) for 30 min, harvested by trypsinisation, and ethanol-fixed. Staining was performed with anti-BrdU primary antibody (1:5 in PBS 1% BSA, BD Biosciences Cat. 347580) and anti-mouse FITC-conjugated secondary antibody (1:50 in PBS 1% BSA, Jackson Immunoresearch Cat. 115-095–075). DNA was stained with overnight incubation with 2.5 μg/ml propidium iodide (Sigma). Samples were acquired on a FACS-CALIBUR (BD Biosciences) flow cytometer and analysed using FlowJo 10 software. Live time-lapse analysis of mitosis and apoptosis during cell cycle re-entry was performed with a high-content screening station (ProScan II, Prior) equipped with a microscope incubation chamber, imaging cells in bright field with a 20 × /NA 0.45 objective (Nikon Eclipse TE2000-E inverted microscope) every 15 min starting at 12 h after serum stimulation and reconstructed with ImageJ software. For each sample, mitosis and cell deaths were manually annotated at each time frame: ~200 cells were followed (in three independent replicas) and data reported as number of mitosis (or cell deaths) occurring every 100 cells.

**Apoptosis analysis by Annexin-V-PI staining**. Cells were grown in normal conditions or under starvation, and treated or not with 1 μM of doxorubicin (Sigma) for 48 h to induce apoptosis. After treatment, adherent as well as detached cells were harvested and labelled with Annexin-V-APC (1:50 in annexin buffer, eBioscience Cat. BMS306APC) for 45 min. Next, cells were labelled with propidium iodide (PI, Sigma) and immediately analysed by MACSQuant Flow cytometer (Miltenyi). Data analysis was performed by FlowJo software (TreeStar).

**Cell viability analysis**. Cells were plated in 96-well plate and analysed at 24, 48, 72, and 96 h by Cell Titre GLO (Promega) following the manufacturer's protocol.

**Isolation of 4sU-RNA**. Starting from 40 μg of total RNA diluted in 100 μl of RNase-free water, 100 μl of biotinylation buffer (2.5 × stock: 25 mM Tris pH 7.4, 2.5 mM EDTA) and 50 μl of EZ-link biotin-HPDP (1 mg/ml in DMF; Pierce/Thermo Scientific 21341) were added and incubated for 3 h at room temperature (RT). Unbound biotin-HPDP was removed by adding chloroform/isoamyl alcohol (24:1) and purifying the mix using MaXtract high-density tubes (Qiagen). RNA was precipitated by adding NaCl (5 M) at 1/10 volume and isopropanol at equal volume. Biotinylated RNA was resuspended in water (55 °C, 5′), quantified with a Nanodrop 1000 spectrophotometer (to keep track of the yield after the first phase of the procedure) and then purified using 50 μl of Dynabeads MyOne Streptavidin T1 (Invitrogen). Before mixing with RNA, beads were washed twice in wash buffer A (100 mM NaOH, 50 mM NaCl) and once in wash buffer B (100 mM NaCl) and then resuspended in 100 μl of buffer C (2 M NaCl, 10 mM Tris pH 7.5, 1 mM EDTA, 0.1% Tween-20) to a final concentration of 5 μg/μl. RNA was added in an equal volume and rotated at RT for 15 min. Beads were washed three times with wash buffer D (1 M NaCl, 5 mM Tris pH 7.5, 0.5 mM EDTA, 0.05% Tween-20). RNA was eluted from the beads in 100 μl of 10 mM EDTA in 95% formamide (65 °C, 10 min). The eluted fraction was diluted in TRIzol® LS Reagent (Life Technologies; cat. no. 10296–028), and RNA was extracted with the miRNeasy Micro Kit (Qiagen; cat.no. 217084) and then eluted in 14 μl of RNase-free water. 4sU-RNA quantification was conducted with Qubit® (Life Technologies).

**Small RNA sequencing and data analysis**. Total RNA, including small species, was isolated through the miRNeasy Mini Kit (Qiagen). When the expected yield was < 1 μg, the miRNeasy Micro Kit (Qiagen) was used. Small RNA sequencing (sRNA-seq) libraries were prepared using 1000 ng of total RNA with the Illumina TruSeq™ Small RNA Kit, following the manufacturer's instructions. Sequencing was performed on an Illumina HiSeq 2000 at 50 bp single-read mode and 20 million read depth (8×). Sequencing quality was checked in the FASTQC report, considering only experiments with Q30 or above (Phred Quality Score). Analysis was performed with the IsomiRage workflow, as previously described[32]. Raw data together with detailed description of the procedures are available in GEO database (GSE104650). Normalisation was performed with the library size (reads-per-million) and then (only for the serum stimulation experiments) further corrected for serum-induced RNA amplification as previously reported[26].

**RNA sequencing and data analysis**. Total RNA was extracted with the miRNeasy Mini (Qiagen) and then 1000 ng were purified with Ribozero rRNA removal kit (Illumina). Libraries were generated with the TruSeq RNA Library Prep Kit v2 (Illumina). Next, sequencing was performed on an Illumina HiSeq 2000 at 50 bp single-read mode and 50 million read depth (3×). Reads were aligned to the mm9 mouse reference genome using the TopHat aligner (version 2.0.6) with default parameters. Differentially expressed genes (244 DEGs shown in Fig. 7a, b) were identified using the Bioconductor package DESeq2 based on read counts, considering genes whose $q$ value relative to the control was lower than 0.05 and whose maximum expression was higher than RPKM of 1. For the identification of genes differentially expressed in MRE-KO cells upon serum stimulation (1648 genes; Fig. 7a, c–f), we compared MRE-KO cells with parental cells at each time-point and we considered all the expressed genes (RPKM > 1) with a log2 FC > |1| in both mutant clones (MRE-KO1 and MRE-KO2) at least at one time-point of the time course. For miRNA-target analysis (shown in Fig. 5), mouse miR-30 targets were downloaded from TargetScan 7.1, including information on conservation and on the type of seed (8-mers, 7-mer-m8, and 7-met-1A). HITS-CLIP–validated miR-30 targets were downloaded from TarBase7.0[35]. CLEAR-CLIP targets were previously published[36]: in particular we focused on the mouse data set, containing brain target:miRNA interactions and we filtered only those interactions occurring in 3′ UTR region.

**miRNA, pri-miRNA and mRNA RT-qPCR expression analyses**. Mature miRNA detection by RT-qPCR was performed with miScript system (Qiagen). In particular, miR-30c-5p and miR-30b-5p were detected using miScript Primer Assay MS00001386 and MS00011725. Primary transcripts of miRNAs (pri-miRNAs) were evaluated by quantitative PCR on total RNA. RT-qPCR was performed using SuperScript® VILO cDNA Synthesis Kit (Life Technologies cat.no. 11754050) and Fast SYBR green master mix (Life Technologies). About 25 ng of cDNA were used as the input to detect pri-miRNAs, and 5 ng to detect mRNAs. Primer pairs were designed using computer-assisted primer design software (Primer3), preferentially in the 500 bp upstream of the sequence of the mature miRNA according to mm10 genome. The complete list of RT-qPCR primers used in this study is in Supplementary Table 1.

**miRNA expression analyses by droplet-digital PCR**. Absolute quantification of miR-30c was performed by droplet-digital PCR on a QX200 system (Bio-Rad), following the manufacturers' instructions. Briefly, 10 ng of RNA was reverse-transcribed with miRCURY LNA Universal RT microRNA PCR kit (Exiqon). Then, ddPCR reactions were prepared replacing Exiqon PCR master mix with QX200 EvaGreen ddPCR Supermix (Bio-Rad) and diluting 1:4 Exiqon PCR primer mix for miR-30c-5p (cat. 204783). Serial dilutions of cDNA (0.4/0.2/0.1/0.05 ng) were used as to measure assay linearity. Droplets were generated and PCR amplification performed following EvaGreen cycling conditions. Finally, we read the droplets on a QX200 droplet reader. We typically obtained ~20,000 droplets for each sample.

**miRNA 3C-target analysis**. Mouse miRNA targets were downloaded from TargetScan 6.2 (i.e., the latest version that includes 3′ pairing contribution scores in downloadable data files). 3C-targets were selected according to criteria outlined in Fig. 1a. Both the conserved sites (CS) and non-conserved sites (NCS) were used. Targets were then classified according to the 3′ end pairing contribution, a feature that measures the presence of additional binding regions (in addition to the seed region) located at the 3′ end of the miRNA. Those targets containing 3′ end supplementary regions are referred to as '3C-target genes' (3C_TG) and classified in three classes (HIGH, MID, LOW) according to the 3′ pairing contribution score, as shown in Fig. 1e. Targets expressed in 3T9 mouse fibroblasts were identified by crossing the list of 3C_TGs with RNA-seq data sets (growing cells and serum-stimulated cells). Genes with FPKM values < 1 were considered as 'not expressed' and, therefore, excluded. Finally, we selected targets of expressed miRNAs in 3T9 cells, by considering sRNA-seq data and filtering miRNA > 10 cpm.

**Generation of Serpine1-MRE30c deletion mutant**. sgRNAs flanking the miR-30 MRE were designed for ENST00000223095.4 (Serpine1 transcript), using the CRISPR Design Tool[34]. SgRNAs were subsequently cloned in PX458 vector and sequence-verified. 3T9 fibroblasts were co-transfected with two PX458 vectors (sgRNA_1 and sgRNA_2) harbouring single sgRNAs sequence (5 μg of DNA for each construct). At 24 h and 48 h post-transfection, cells were harvested for FACS sorting. The top 2%–3% of GFP + cells were sorted as individual cells into 96-well plates. After 2 weeks, 96-well plates were duplicated and deletions in clonal cells were screened by PCR (Supplementary Fig. 4a) using specific primer to detect mono-allelic or bi-allelic deletions. Positive clones were confirmed by DNA sequencing (Supplementary Fig. 4b). The protocol efficiency was about 20% for mono-allelic and 2%–5% for bi-allelic deletions. PCR primers for clone screening were Serpine1_crisp_FW: GGAGGGCACAACACTTTCAT; and Serpine1_crisp RW: AGTGCTTCTTCTCCCCAACA. MRE-KO clones used in this work were produced by the following sgRNAs:

 sgRNA_1(-) Sense: caccgAAGCAAGCTGTGTCAAGGGA
 Antisense: aaacTCCCTTGACACAGCTTGCTTc

 sgRNA_2 ( + ) Sense: caccgTCTCCCAGTGGGGGGGCCCT
 Antisense: aaacAGGGCCCCCCCACTGGGAGAc
Note that the efficient transcription from the U6 promoter requires a 5′ g.
Serpine1 protein expression in MRE-KO clones was analysed by western blot (PAI-1 monoclonal antibody, 1D5, dilution 1:1000, Thermo Fisher Cat. MA5-17171).

**Rescue of MRE expression by adenoviral constructs**. In order to re-express miR-30 MRE, recombinant adenoviruses were produced using the AdEasy System (Agilent), following manufacturers' instructions. Briefly, gene synthesis (GenScript) generated a mCherry-Serpine1 (3′UTR) construct (see below for the complete sequence) that was cloned into pShuttle vector (Agilent) by adding KpnI-NotI sites at the extremities. After homologous recombination in BJ5183-AD-1 cells (Agilent), positive recombinants (isolated by PacI digestion) were used to produce adenoviruses in 239Ad cells. Viruses were finally concentrated with Adeno-X maxi purification kit (Takara).

mCherry-Serpine 3′UTR construct (underlined with a solid line the miR-30 MRE):
5′-
GGTACCCGCCACCATGGTGAGCAAGGGCGAGGAGGATAACATGGCCAT
CATCAAGGAGTTCATGCGCTTCAAGGTGCACATGGAGGGCTCCGTGAAC
GGCCACGAGTTCGAGATCGAGGGCGAGGGCGAGGGCCGCCCCTACGAG
GGCACCCAGACCGCCAAGCTGAAGGTGACCAAGGGTGGCCCCCTGCCCT
TCGCCTGGGACATCCTGTCCCCTCAGTTCATGTACGGCTCCAAGGCCTAC
GTGAAGCACCCCGCCGACATCCCCGACTACTTGAAGCTGTCCTTCCCCG
AGGGCTTCAAGTGGGAGCGCGTGATGAACTTCGAGGACGGCGGCGTGG
TGACCGTGACCCAGGACTCCTCCCTGCAGGACGGCGAGTTCATCTACAA
GGTGAAGCTGCGCGGCACCAACTTCCCCTCCGACGGCCCCGTAATGCAG
AAGAAGACCATGGGCTGGGAGGCCTCCTCCGAGCGGATGTACCCCGAG
GACGGCGCCCTGAAGGGCGAGATCAAGCAGAGGCTGAAGCTGAAGGAC
GGCGGCCACTACGACGCTGAGGTCAAGACCACCTACAAGGCCAAGAAG
CCCGTGCAGCTGCCCGGCGCCTACAACGTCAACATCAAGTTGGACATCA
CCTCCCACAACGAGGACTACACCATCGTGGAACAGTACGAACGCGCCGA
GGGCCGCCACTCCACCGGCGGCATGGACGAGCTGTACAAGTAGCTCGAG
CAGTGGGAAGAGACGCCTTCATTTGGGACGAAACTGGAGATGTTATAAG
CAGAAACTCTGAAGAAAAGGATTATTTAAAGGACTCTATGGGGAGAAAG
AGAAGGCAACTCCTCCTTACCCCCCACACTGGTAATCTTTCCAACCAGCA
TCCCAGACCTCGGACTCTTGAAGGGAAAAGAGTCTAACTCCCTCCTTCCC
TAGGGATTCCTACCCCACAAAGGTCTCATGGACCATAGAACTCACAGTA
CCTGGATCTGCCCAGCATGCCCTTTGGACCCAGTTCCCACCGAGGCCCC
AGCAGAGTGGAGGGCACAACACTTTCATTCAGCAAAATCGTTTGTGTTC
CAGTCACACTGTGGGCACCTCTTGCATCGCCTGCCATTGCTGTGGAGGG
TGGCCATGGGCCAAAGGAAAAAACACTGTCCTATCTCAAGGTCCACTGT
GGAAATGTCCACCTTGCCCACCTCCAAGGGGCAACGGATAGACAGATCA
AATGGTGGCCCAATAGCGAGCCTTCTCCCTGCTCCCTCCCTTGACACAGC
<u>TTGCTTATGTTATTTCACGCGTAGAGTGTAGGTGACTTGTTTACAGAGC
TCC</u>AGCTTTTTTCGA<u>CCCACAAACTTTTTTCATTTGGAAAGGGTGTAAGA
AAA</u>GTCGGACGTGTGTGTGCCTGGCTCTTCGTCCCCAGTCTCCCAGTGG
GGGGGCCCTGGGGAGATTCCAGGGGTGTGATTGAATATTTATCTCTTGC
TCTTGTATGTTTGTTGGGGAGAAGAAGCACTTTTAAGGAAAATGCTTCT
TATTTAAACCGTGGCATACGGCATCCCATTTGGGGTCTGCATCCCTGTAT
GTCAGGGGTGCATCACTCCACAAACCTGCCCCTCTGGGTAGCCTCGTGA
TGGGGCTCACACTGCCGCCTAGTGGCAGCCGAACACACCCTTACCCGGT
CCCTCCCTCCCTCCCCCCCCCCCCCCCCCCCCCGTGGCTCTTTTCCTT
AGGGACCTTGCCAAGGTGATGCTTGGCAACCCACGTTAAAGGAAGGGGG
GAAAAAAGATTAGATGGAAGAGAGAGAGAGATTTGAGAGAGGGCAAAGTG
GTTTCAAATTTTTCCAAGGCATTCAGAAGCAGAGAGGGAAAAGGGGCTG
TGTGACCTAACAGGACAGAACTTTCTCCAATTACTGGGTGAGTCAGAGC
TGCACTGGTGACTCACTTCAATGTGTCATTTCCGGCTGCTGTATGTGAG
CAGTGGACACGTGGGGGGGCGGGGGGGGGGATGAAAGAGACAGCAGCTC
CTGGTCAACCACCTTAGTTAGATAATCTTTTTTGAAAGCTTCCTAGCTGG
AGGTATGATCAGAAAACCAATTTACTGAAAAACTGCACAAGAAGGTACG
GTGAATGTAATTTCCTAGCAGGCCACTCTGCATCTGTTATGTCTCCACCG
GAAAAAAAATAATCATGTTGGTGTTTTGCTTTTCTCTCTCCCTCTTTC
TCTCTGATTTTTTTTTCCTCTCTTTTCATTATGCACTGGACAGCCACACAC
CGTGTACCCATAGGGCCCCAAATGTGGGGTCACATGGTCTTGAATTTTG
TTGGTTACATATGCCTTTTTGTTGTTGTTTGTCTTCACTTTTGATATATAA
ACAGGTAAATATGTTTTTTAAAAAAATACTAAATATAGAGAATATGCAAA
CAAAAGCGGCCGC-3′
 GGTACC KpnI
 ACGCGT MluI
 GAGCTC SacI
 GCGGCCGC Not1
In order to produce other adenoviral constructs, the following oligonucleotides were subcloned into MluI-SacI sites flanking the miR-30 MRE:
 Ad-RFP-SE1(MUT)
 FW: cgcgtACTCTGAAAGTGAAAAGCCAATGgagct
 REV: cCATTGGCTTTTCACTTTCAGAGTa
 Ad-RFP-SE1(SEED)
 FW: cgcgtGATATTGTCCTGACTTGTTTACAgagct
 REV: cTGTAAACAAGTCAGGACAATATCa

**Expression of mutant-binding sites for miR-30**. In order to re-express different variants of Serpine1$^{miR-30\_MRE}$ in MRE-KO cells, we cloned MRE + ~20 surrounding bases (Supplementary Table 2) into *Eco*RI-*Xho*I sites that were put after the stop codon of a eGFP cloned in a pcDNA3.1( + ) mammalian expression vector (Thermo Fisher Scientific).

Vectors were transfected in MRE-KO cells with Lipofectamine 3000 reagent (Thermo Fisher Scientific). Cells ( > 75% GFP positive by FACS analysis) were collected for analysis after 48 h.

**Measurement of miR-30 activity (miR-sensor) by flow cytometry**. Sequences harbouring four complementary repeats for miR-30c-5p or a control sequence (see below) were purchased from TwinHelix and cloned (*Xba*I/*Xma*I) into the 3′UTR of a destabilised GFP (dGFP) in a bidirectional lentiviral vector (BdLV_1370), which also expresses ΔNGFR. The vector (schematised in Supplementary Fig. 6) was kindly provided by Luigi Naldini and his collaborators. Infected cells were analysed by FACS to isolate populations with similar ΔNGFR expression as previously[41]. Samples were analysed by counting 100,000 events per replica using a MACSQuant Flow cytometer (Miltenyi). Data analysis was performed by FlowJo software (TreeStar): fluorescent levels of ΔNGFR and dGFP for each cell were log transformed and divided in 100 bins (each of 1000 cells) based on ΔNGFR level. The mean ΔNGFR or dGFP expression was calculated in each bin and repression folds were calculated, at similar ΔNGFR mean expression, as the ratio: [dGFP(miR-30)/[dGFP(control)].

miR-30c-5p sensor sequence:
TCTAGATAAGCTGAGAGTGTAGGATGTTTACACGATGCTGAGAGTGT
AGGATGTTTACAACGCGTGTCGACGCTGAGAGTGTAGGATGTTTACATC
ACGCTGAGAGTGTAGGATGTTTACACCCGGG

Control (scrambled) sensor sequence
FW TCTAGAGGAGCTCCACCGCGGTGGCATC
REV CCGGGATGCCACCGCGGTGGAGCTCCT

**Data availability**. The small RNA sequencing (sRNA-seq) and the RNA-sequencing data set for this study have been submitted to the NCBI Gene Expression Omnibus (GEO; http://www.ncbi. nlm.nih.gov/geo/) under the accession number GSE104650. The 'serum stimulation data set', which contains miRNA expression analysis of WT cells, was previously submitted under accession number GSE72655. Data within the manuscript is available from the authors upon reasonable request.

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

## Acknowledgements

We thank the Genomic Unit (Luca Rotta, Salvatore Bianchi, Thelma Capra) for sequencing runs and Claudia Crovace for manuscript editing. This work was supported by grants from the Associazione Italiana per la Ricerca sul Cancro (AIRC) to F.N. (IG14085 and IG18774), Cariplo (2015-0590) to F.N and Cariplo (2016-0615) to M.J.M.

## Author contributions

F.G. generated MRE-KO cells. F.G. and C.R performed characterisation of precursor and mature miRNAs, generated adenoviruses and performed reconstitution experiments. F.G., C.R., M.C. performed biological characterisation of MRE-KO cells. Samples for

sequencing were prepared by M.J.M. and F.G., F.N., M.J.M. and I.S. performed bioinformatics and statistical analyses. M.J.M. and F.N. analysed data and wrote the manuscript.

## Additional information

**Competing interests:** The authors declare no competing interests.

