## [Peer Review File · Nature Communications]

Reviewers' comments:

Reviewer #1 (Remarks to the Author):

This manuscript by Ghini and colleagues describes the interesting finding of the first example of an endogenous mRNA target (Serpine) that can control the levels of the endogenous miRNA (miR-30b/c-5p) by inducing miRNA degradation. While high profile papers have described a similar phenomenon of target-dependent miRNA degradation (TDMD), this work is the first to report evidence that this can occur for endogenous mRNA/miRNA pairs.

Overall the results are quite convincing and the experiments done to a high standard. In particular results from the CRISPR KO of the MRE are particularly striking. With some modification this manuscript should be suitable for publication in Nature communications.

Major Points

1) A key feature of the proposed TDMD mechanism is the 3' complementarity. However, the data demonstrating the necessity of 3' comp is largely inadequate. Can the authors perform a more thorough examination of the necessity of the 3' comp? E.g. for Fig 4A, besides WT and SEED, generate constructs with "low", "med" and "high" 3' complementarity and check for their effects in comparison to WT and seed constructs.

2) From Figure 2B, it is very clear that Serpine1 contributes only negligibly to the pool of all seed only targets, though it seems that Serpine1 contributes significantly to the 3C high pool. What proportion of the miRNAs are expected to reside on the "seed only" sites vs. the 3C high sites? Indeed, if the majority of the miRNAs reside on the seed only sites, it seems unlikely that the miRNAs can be effectively sequestered by Serpine1 mRNA. Can the authors check if there are any relevant CLASH datasets from which they can determine the relative occupancy of a miRNA towards seed only, low, mid and high sites?

3) Figure 1E: The "% of 3C pool" was calculated as: RPKM target / RPKM Sum_all_3C-targets. As multiple time points were used with the RPKM of the target and all 3C targets most likely varies considerably with respect to each other, the percentage contribution would most likely differ depending on the time point utilized. Which time point was used in this case and why was it chosen?

4) Figure 3H. What time-point was the analysis performed at? Was it performed on GW, 0h, 2h or 4h samples? If the analysis was performed on GW/0h samples, the levels of Serpine1 is presumably too low to induce adenylation and this phenomenon cannot be directly attributed to serpine1 MRE?

5) Figure 4C,D - A titration was performed for WT and Mut but not the "seed" form. Can the authors perform the titration experiment for the "seed" form as well?

Minor points:

1) Introduction (page 3). The phrase "Due to the low levels of complementarity between miRNAs and RNA targets, hundreds to thousands of RNAs could interact with a single miRNA" seems somehow misleading. Perhaps changing to a "...single miRNA sequence" since it is not an individual miRNA molecule that interacts with thousands of mRNAs (i.e. there should be a 1:1 stoichiometry)

2) Figure 1C: It is unclear (and not described in the legend) what the box and whisker plots show.

3) Figure 1 legend – Typos: "(C) Distribution plots of the fraction of 3C-targetss" and "(D) 3C-targetss" should be "targets".

4) Page 10 – Typo: “The RFP-SE1-WT construct significantly reduced the levels of miR-30c and, to less extent, miR-30b” should be “.., “to a lesser extent..”

5) Figure 1E legend. "High" is represented by score " < -0.05 " and "Low" is represented by score " > -0.03 "?? Why is the directionality switched? It makes it somewhat confusing. (A more negative value should be lower than a less negative value).

6) Beside Serpine1, what were the other miR-30b/c targets? What is their percentage contribution towards the 3C pool at different time points of the study?

7) What was the adaptor used in the small RNA sequencing experiment? An improperly stripped adaptor sequence could falsely contribute towards the profile observed in Figure 2F. At the same time, this could also create the impression of a "trimmed miRNA".

8) For Figure 2F, were some of these non-canonical tailings a result of sequencing errors? What was the base quality cutoffs ($> Q30$??) used when generating this figure?

9) From Figure 3F, it seems clear that there were a number of miRNAs that up/down regulated upon MRE-KO. What are the identities of these miRNAs and were these changes related to the TDMD mechanism the authors proposed?

10) What does the coloring of the points in Fig 3F represent?

11) Figure 4B- 30b-5p and 30e-5p results look fairly similar but the authors labeled one as $p=0.059$ and the other as "n.s". What is the p-value for 30e-5p?

12) Figure 4D - results for 30b-5p is missing.

13) Figure 4F. Tailing results for 30a/d/e are missing.

14) Figure 5D. The color of the lines (red + blue) differ from the labels which are in green and gray.

Reviewer #2 (Remarks to the Author):

The Target-Dependent miRNA Degradation (TDMD) is a relatively recently described phenomenon. Its principles were established with the use of artificial targets and it remained unknown whether TDMD indeed operates in more physiological settings and is induced by natural miRNA targets. Ghini et al. provide strong evidence that Serpin1 mRNA may act as TDMD inducer. Several lines of evidence support their conclusions. The work is generally carefully done with the use of state-of-the-art assays, also providing indications of the biological effect of miR-30b/c degradation on apoptosis and cell cycle.

Identification of the natural inducer of TDMD is certainly a very important finding which will greatly stimulate research in the field.

A few problems, however still remain and they will need to be addressed before the paper is considered for publication.

Major points:

1. Although evidence that miR-30b/c but not other family members (miR-30a/d/e) are targeted for degradation by Serpin1 is quite strong and strongly suggests that extensive complementarity at the miRNA 3'-end is required for degradation to occur, I find the effect of about 2-fold decrease of miR-30b/c on endogenous targets and cell behavior rather surprising. Based on cpm data in Fig.

2C (number of RNA Seq reads), miR-30b/c makes only ~10% of the total amount of the family. Hence, after TDMD of miR-30b/c, 95% of the total complement of the miR-30 family remains intact (all family members share the seed).

Why then such a strong effect on endogenous targets and on cells. The authors do not discuss this issue at all. Are miR-30 variants with less of the 3' complementarity so much less effective in miRNA repression? This could be addressed experimentally by testing them individually with one or two artificial reporters.

2. It is not clear, particularly for the second part of the results, starting with Fig 3 and further, under what conditions of cell growth are all the assays and quantifications done. Are the authors always looking at cells treated with serum to induce Serpine1 expression? If so, at what hr post-induction?

In cells grown under control conditions, the level of Serpin1 mRNA is very low (see, for example, Fig. 4C). At this level of Serpin1 mRNA, one would not expect a particularly strong effect on the miR30c/b level (despite the fact that small ~2-fold difference is observed in corresponding figure). Hence, the difference between WT and MRE-KO cells shown in Fig.5B (5.5 versus 7.7-fold repression of the reporter) is rather surprising. Same question applies to other panels in Fig.5 and the following figures.

The aforementioned confusion extends to discussion. On p. 15, line 9 bottom, it is said that target mRNA (Serpine) and miR30b/c are present at ~200 cpc. This is not consistent with data in Fig. 4C/D.

3. Previous studies reported quite different requirements for the 3' end complementarity in order to induce TDMD. While Amares et al. found that even 5-6 non-complimentary 3'-terminal pairs are tolerated, de la Mata et al. found very strong effect of even two mismatches. The authors' data suggest that the miR-30 /Serpine TDMD tolerates at least 4 mismatches. This issue needs to be discussed.

4. The authors investigated before, globally, turnover of all miRNAs expressed in fibroblasts, putting them into different decay categories. To what category were the miR-30 miRNAs classified?

5. In Fig. 4C/D, expression of the seed mutant is very low, perhaps too low to see any effect. Why only single MOI is used for this important mutant? Could higher MOI be tested to make this control stronger?

6. In Fig. 4E, in contrary to the text, the level of miR-306 is changed. This issue and potential explanation is discussed in Discussion. This observation needs also to be mentioned on p.10 to avoid confusion.

7. Figure 4G. It is not clear how the levels of SE1 WT reporter were varied to obtain the presented correlation. A battery of different MOIs was used?

Minor points:

1. P.10, l. 9 top. Rather: "... Serpine 1 constructs" (plural)

2. P. 7, l. 3 bottom. Rather "... remained ALMOSTunchanged...." Some effects are seen

3. P. 11, l. 5 top. Sed (?) fluorescence?

4. P. 16, l. 6 bottom. Referring to ceRNA-like effect is not a good idea. CeRNA was never documented to involve miRNA degradation.

Reviewer #3 (Remarks to the Author):

RNAs can induce degradation of miRNAs if they bind them with extensive complementarity in a process also known as TDMD. However, endogenous TDMD-inducing RNAs have remained elusive. Ghini et al. identify Serpine1 mRNA as one such RNA, acting on miR-30b & c. They further report that this regulation is biologically relevant in serum-stimulated cells, where failure to reduce miR-30b, c levels causes inappropriate target silencing, and ultimately cell death.

I find the identification and characterization of Serpine1 as a TDMD-inducer convincing and interesting. I am less convinced by the deduced biological implications as detailed below. The

authors should address the following issues:

Major points

1. The authors claim that *Serpine1* mRNA specifically initiates TDMD on miR-30b/c but not three other miR-30 family members. Loss of this *Serpine1*-mediated TDMD (through MRE knock-out) causes specifically upregulation of these two family members, which then results in increased silencing of miR-30 family targets, and cell death. This scenario is unconvincing, because miR-30b and c appear to account only for a very small fraction of total miR-30 family miRNA molecules in the cell (cf. Fig. 2C). This could be a consequence of sequencing biases, which should be ruled out by absolute quantification of all family members. However, if, as likely, true, it is unclear how such a modest change of overall miR-30 level would then decrease target levels. Indeed, although the data indicate a ratio of one for miR-30c to *Serpine1* mRNA levels (p. 8, para 2), this ratio would be much diminished by competing *Serpine1* binding through other family members.

2. Consistent with the concern about miRNA family member levels, although Fig. 5D shows a statistically significant increase in target silencing, the differences in predicted target level repression in MRE-KO vs. wt appear minute (5%). Even with the modest effects we have come to accept in the miRNA field, it is hard to imagine how this would result in a biological effect. Authors should show directly that a <2-fold over-expression of miR30b/c (during continued expression of other miR-30 family members) recapitulates the effects of MRE-ko on both targets and cell death/cell cycle (below).

3. In the same vein, the 'miR-sensor' used in Fig. 5B contains 4 sites of perfect complementarity to miR-30c. This would immediately bias the read-out to miR-30c, away from other family members, although the sensor is claimed (p.10, last para) to reflect miR-30 (family) activity. I would agree that sensing changes of the entire miR-30 family is what needs to be done when testing the authors' model, but for this they should use a seed match, not a perfect match, target reporter. Alternatively, if the authors have identified endogenous targets of miR-30c that are relevant for the phenotypes, and specifically only regulated by miR-30c but not by other family members, they should investigate these, endogenously or at the level of a derived reporter. This would reflect a different hypothesis from that stated on p. 10 though. Incidentally, repression did not increase by 7.7-fold but to 7.7-fold in the MRE-KO (description of Fig. 5B, p.11, para 1).

4. In Fig. 6B, the authors imply that "morbidity or mortality", "organismal death" and "perinatal death" are all aspects of "cell death" – I consider this a leap and most likely wrong, since they all refer to organismal not cellular level. Accordingly, although MRE-KO cells appear to be more prone to cell death under stress conditions, the evidence that this is through "over-repression" of miR-30b/c targets is weak. Again, recapitulation of these phenotypes by modest miR-30b/c over-expression (point #2 above) would at least provide some circumstantial support for the model, although ideally there would be some data to implicate a specific set of targets.

Additional points:

1. Fig. 1E, p.7, para 1: Please define the "max contribution of each single target to the total pool of 3C-targets of a given miRNA" – is this based on number of predicted targets overall, above the expression cutoff as defined in Fig. 1D, or weighted by abundance? I assume the latter, but it is not clear. A key to convert bubble size to target abundance would also be helpful.

2. To validate that miR-30b/c expression level changes occur at the level of the guide strand, please provide from the sequencing data a plot of passenger strands for all five family members over time.

3. Fig. 3D/E: *Serpine* mRNA and protein levels appear unchanged or even down in the MRE mutant. This seems surprising because it ought to be a target of the whole miR-30 family so that

loss of MREs would be expected to cause upregulation. Please explain.

4. Fig. 4D,E: Why does miR-30b-5p remain unchanged by reintroduction of wt-Serpine, and what would the corresponding graph for Fig. 4D look for it?

5. Fig. 4F: Why are n's all different for the three transgenes, have data been omitted? If so, please justify.

6. Fig. 4F vs. 3H: tailing levels appear globally reduced (irrespective of the type of transgene added) in 4F vs. what is reported for the ko cells in 3H. This difference is much bigger than the effect that results from introducing wt-Serpine1, making me wonder how robust this assay is – please comment.

7. P.10, para 2/ Fig. 4G/H: The authors argue that they require much higher levels of Serpine1 mRNA to elicit degradation when transfected than when endogenously present because there is only modest transduction efficiency at the low value. I can accept this argument. However, Ghini et al. then use the same experiment to argue that Serpine1 mRNA has a dose-dependent, not a thresholded, function on miR-30c-5p levels. Clearly, if the preceding explanation were true, this experiment would be unsuitable to provide evidence for or against a threshold mechanism, so I cannot accept their conclusion.

Minor points

1. Phrasing tends to be at times vague or misleading, please carefully review. A few examples are below:

a) p.3, para1: "...however, other sequences, such as compensatory sites located..." – as it is not clear, what would be "compensated for", the authors should use "complementary" not "compensatory", and do so consistently throughout the manuscript. "Sites" is also misleading since the miRNA binding site would be composed of stretches of seed complementarity and complementarity to the 3'end.

b) *ibid*: "...hundreds to thousands of RNAs could interact with a single miRNA, often simultaneously..." – this reads as if one molecule of miRNA can silence many targets in parallel, but I assume what the authors want to say is that within a cell type, miRNAs can have many different targets?

c) *ibid*: "The interaction between miRNAs and targets occurs at the level of Argonaute proteins..." – what does this mean?

d) The authors refer to De et al. 2013, de la Mata et al. 2015, and Park et al. 2017 as "in vitro" studies, p.4, p.6, which is a bit confusing considering that two of the studies were performed in a test tube, the third one on cells. More clarity would be useful, also in assigning an appropriate label to their own study.

2. Krol et al. Nat Rev Genet 2010 is cited as a review on TDMD, but does not treat the topic; an appropriate reference should be given. Item, Krol et al Cell 2010 does not report on TDMD.

3. Fig. 1A: define "NCS" and "CS" in figure legend.

4. Fig. 1D: how were the thresholds for expression filtering selected?

5. The discussion should be more concise and focused, e.g. the question of target vs. miRNA stoichiometry is important does not need to be discussed twice (p.14, p.17).

Response to Reviewers' comments:

Reviewer #1 (Remarks to the Author):

This manuscript by Ghini and colleagues describes the interesting finding of the first example of an endogenous mRNA target (Serpine) that can control the levels of the endogenous miRNA (miR-30b/c-5p) by inducing miRNA degradation. While high profile papers have described a similar phenomenon of target-dependent miRNA degradation (TDMD), this work is the first to report evidence that this can occur for endogenous mRNA/miRNA pairs.

Overall the results are quite convincing and the experiments done to a high standard. In particular results from the CRISPR KO of the MRE are particularly striking. With some modification this manuscript should be suitable for publication in Nature communications.

Major Points

*1.1) A key feature of the proposed TDMD mechanism is the 3' complementarity. However, the data demonstrating the necessity of 3' comp is largely inadequate. Can the authors perform a more thorough examination of the necessity of the 3' comp? E.g. for **Fig 4A**, besides WT and SEED, generate constructs with "low", "med" and "high" 3' complementarity and check for their effects in comparison to WT and seed constructs.*

R. We thank the reviewer for the comment and agree that our reconstitution experiment demonstrated the need for a 3' complementarity, but did not investigate further the degree of complementarity required for TDMD. With regard to ectopic expression, pairing requirements at the miRNA 3' end were largely analyzed in previous papers (Ameres et al. Science 2010, De la Mata et al EMBO Reports 2015), leading to the conclusion that i) extensive but not complete pairing is required and ii) reducing the pairing (even 1 nucleotide) strongly impacts on TDMD effectiveness. In agreement with these conclusions, we showed that the different levels of complementarity between Serpine1 and the five members of miR-30 family resulted in different degrees of sensitivity to target-induced degradation. The most potent TDMD effect was observed for miR-30c, which has a match of 11 (10 consecutive) nucleotides (**Fig. 2a**, originally indicating a match of 9 nt due to a typo, has been amended), while miR-30b (which has two additional 3' mismatches) was still affected, even if to a minor extent (see **Fig. 3g**). The preference for miR-30c became more evident when, in the reconstitution experiments, Serpine1 MRE had been ectopically expressed (see **Fig. 4b,e**). Differently, lacking of extensive complementarity, miR-30a/d/e were insensitive to Serpine1 induced degradation. Nevertheless, to further characterize Serpine1 pairing requirements, we generated a set of mutants with different levels of complementarity in the 3' region (see **Fig. 4g,h**). Our results showed that pairing of 8 nucleotides is needed for TDMD to occur and mismatches can be tolerated if they are close to the bulge, while strongly impairing miRNA degradation when located in proximity of miRNA 3' end. This new piece of information has now been included in the revised manuscript (page 11, lines 223-227, and page 14, lines 322-324).

*1.2) From **Figure 2B**, it is very clear that Serpine1 contributes only negligibly to the pool of all seed only targets, though it seems that Serpine1 contributes significantly to the 3C high pool. What proportion of the miRNAs are expected to reside on the "seed only" sites vs. the 3C high sites? Indeed, if the majority of the miRNAs reside on the seed only sites, it seems unlikely that the miRNAs can be effectively sequestered by Serpine1 mRNA. Can the authors check if there are any relevant CLASH datasets from which they can determine the relatively occupancy of a miRNA towards seed only, low, mid and high sites?*

R. As suggested, we looked for datasets mapping miRNA:target interaction based on biochemical approaches. We retrieved two relevant works. In the first paper (Helkaw A et al. Cell 2013 – David

Tollervey Lab), the authors used CLASH and showed that separate classes of miRNAs interact differently with their target RNAs, as in the case of some miRNAs presenting a match in the miRNA 3' region. In this particular class of miRNAs is also included miR-30c (Figure 3D of the Helkawa paper), suggesting a different target specificity for miR-30b/c as opposed to miR-30a/d/e. In the second paper, the authors used a modified HITS-CLIP strategy with covalent ligation of Argonaute-bound RNAs (CLEAR-CLIP, Moore MJ et al. Nat Comm 2015 – Bob Darnell lab) and mapped endogenous miRNA:target interactions in mouse brain and human hepatoma. The authors showed that some miRNAs, including miR-30c and to a lesser extent -30b, use auxiliary pairing (the seed plus a 3' region) to interact with their targets (Figures 6d and 7c of the Moore MJ manuscript) and included the MiR-30c:Serpine1 in their mouse database (Suppl. Table 1 of the CLEAR-CLIP paper). Of note, Serpine1 was scored among the preferred miR-30c interactions (according to the predicted minimum free energy for duplex structure – MFE), being in the top 2% of total found interactions (n. 41 of 3569) and in the top 0.25% of 3'UTR interactions (n.2/935) for miR-30c. Based on these observations, it is reasonable to conclude that miR-30c preferentially binds to those targets that possess a supplementary pairing in the 3' region (3C targets). With regard to the mouse fibroblast model, Serpine1 contributes significantly to the miR-30c target pool (it represents up to 20-30% of the 3C Target Pool, as pointed out in **Fig. 2b**). We thank the reviewer for their comment as this important piece of information has been now included in the revised manuscript (page 11, lines 244-246, and page 12, lines 259-262). In addition, we utilized CLEAR-CLIP data to gain an insight into the regulation of miR-30 targets upon TDMD disruption (included in **Fig. 5h,i**) and to investigate the specific subset of miR-30b/c targets (**Fig. 5j,k**).

1.3) Figure 1E: *The "% of 3C pool" was calculated as: $RPKM\ target / RPKM\ Sum_all_3C\ targets$. As multiple time points were used with the RPKM of the target and all 3C targets most likely varies considerably with respect to each other, the percentage contribution would most likely differ depending on the time point utilized. Which time point was used in this case and why was it chosen?*

R. **Fig. 1e** showed the maximum contribution over the entire time course, rather than at a specific time-point. The aim was to explore the entire time course to isolate those targets that, in any time point, contributed significantly to the 3C_target pool. In the case of Serpine1:miR-30c interaction (highlighted in **Fig. 1e**), the max contribution was reached at 2 hours after serum stimulation (where Serpine1 contribute to ~35% of the miR-30c 3C-target pool). **Figure 1 legend** has been revised accordingly

1.4) Figure 3H. *What time-point was the analysis performed at? Was it performed on GW, 0h, 2h or 4h samples? If the analysis was performed on GW/0h samples, the levels of Serpine1 is presumably too low to induce adenylation and this phenomenon cannot be directly attributed to serpine1 MRE?*

R. We thank the reviewer for their comment. **Fig. 3** has now been updated to show in panels **g-h-i** the characterization of the miRNA landscape in exponentially growing fibroblasts (GW) from WT and MRE-KO clones. Panels **j-k-l** show the characterization at 0h and during the time course of serum stimulation. We agree that in GW/0h samples Serpine1 levels are relatively low, in particular if compared with the massive induction at 2 hours upon serum stimulation (see **Fig. 3a**); however, we have to point out that target-dependent degradation is an irreversible process, which, in principle, is not entirely dependent on target levels. As suggested by Denzler et al. (Bartel lab -Mol Cell 2016), in order to induce significant miRNA degradation, the target does not need to reach the effective abundance of all miRNA target sites, as every time the miRNA interacts with the target it is irreversibly subtracted from the miRNA pool by degradation ('multiple turnover activity'; de la Mata, et al., EMBO Rep 2015). The level of miRNA will decrease to reach a new equilibrium, where the miRNA levels are lower and those of the degradation intermediates are increased. Therefore, even targets expressed at low levels can influence miRNA decay. Of course, the higher the target expression, the bigger the change in miRNA levels, and the faster the kinetic of the process. In **Fig. 3g-i** we measured miRNA levels in growing cells at steady-state, which was

reached after several days of culture (almost four weeks were needed for clone expansion and characterization). The effects observed on miR-30s are totally compatible with this model, as Serpine1 is expressed at relatively low levels (~100 cpc) but still in a stoichiometric range with miR-30c (about 70 cpc). During serum stimulation, Serpine1 is expressed at extremely high levels (up 4000 cpc) for a few hours and miR-30c degradation is thus accelerated in a precise time frame. Regarding miR-30b/c adenylation, the reviewer suggests that the increase in adenylation (described in **Fig. 3i**) could not be directly attributed to Serpine1. Changes in adenylation of miR-30b/c have been consistently linked to Serpine1:miR-30b/c interaction in multiple independent experiments, including i) the rescue experiments, where miR-30c levels go down and adenylation increases following re-expression of Serpine1 MRE (**Fig. 4f**); ii) during serum stimulation (**Fig. 2d, Fig. 3k**), when Serpine1 is massively induced, miR-30c/b levels go down and adenylation increases dramatically (see also **Supplementary Fig. 5 – WT**), a phenomenon that is specifically abrogated in MRE-KO cells (**Supplementary Fig. 5 – KO**). As previously said, we updated **Fig. 3**, distinguishing data obtained in growing conditions from those of the time course of serum stimulation and updated discussion including comments on the quantitative requirements of TDMD (page **15**, lines **325-342**).

1.5) Figure 4C,D - A titration was performed for WT and Mut but not the "seed" form. Can the authors perform the titration experiment for the "seed" form as well?

R. We thank the reviewer for their comment. We performed additional experiments to express the 'seed' mutant at different doses, in line with WT and MUT constructs. Results confirm that the seed-match is not sufficient for degradation to occur even at extremely high doses (>10,000 cpc). We amended figures accordingly (**Fig. 4c,d**).

Minor points:

1) **Introduction (page 3)**. The phrase "Due to the low levels of complementarity between miRNAs and RNA targets, hundreds to thousands of RNAs could interact with a single miRNA" seems somehow misleading. Perhaps changing to a "...single miRNA sequence" since it is not an individual miRNA molecule that interacts with thousands of mRNAs (i.e. there should be a 1:1 stoichiometry)

2) **Figure 1C**: It is unclear (and not described in the legend) what the box and whisker plots show.

3) **Figure 1 legend** – Typos: "(C) Distribution plots of the fraction of 3C-targetss" and "(D) 3C-targetss" should be "targets".

4) **Page 10** – Typo: "The RFP-SE1-WT construct significantly reduced the levels of miR-30c and, to less extent, miR-30b" should be "...", "to a lesser extent.."

R. We thank the reviewer for the comments. We have amended text and figure legend accordingly.

5) **Figure 1E legend**. "High" is represented by score "< -0.05" and "Low" is represented by score "> -.03"?? Why is the directionality switched? It makes it somewhat confusing. (A more negative value should be lower than a less negative value).

R. The 3C score is expressed as a negative value, calculated by TargetScan and taking into consideration the greatest number of contiguously paired bases, weighted towards pairing nucleotides 13-16 (as described in Grimson et al. Mol Cell 2007). Therefore, the higher the complementarity the lower the score. Based on this premises, the 'High' class, which has the greater complementarity, scores below -0.05. To clarify this issue, we amended the figure legend (**Fig. 1e**) and the text in the revised manuscript (page **6**, line **100**).

6) **Beside Serpine1**, what were the other miR-30b/c targets? What is their percentage contribution towards the 3C pool at different time points of the study?

R. We thank the reviewer for their comment. In 3T9 cells, miR-30b/c has 225 3C-targets, of which 16 are in the 'High' ($3C < -.05$) complementarity group. We followed these 16 targets over the time course, but none of them contributes significantly to the total target pool at any time point (the max is Pls3 with a 2.34% max contribution). This piece of information has been added as **Supplementary Fig. 2a**.

7) *What was the adaptor used in the small RNA sequencing experiment? An improperly stripped adaptor sequence could falsely contribute towards the profile observed in **Figure 2F**. At the same time, this could also create the impression of a "trimmed miRNA".*

8) *For **Figure 2F**, were some of these non-canonical tailings a result of sequencing errors? What was the base quality cutoffs ($> Q30$??) used when generating this figure?*

R. We thank the reviewer for their comment. Our group has previously developed a pipeline for small RNA sequencing experiment analysis, named IsomiRage (Muller et al Front. Bioeng. Biotechnol. 2014), tailored to the analysis of miRNA isoforms. As described in methods, libraries were prepared with the Illumina TruSeq™ Small RNA kit. The CutAdapt tool was used for adapter removal (TGGAATTCTCGGGTGCCAAGGAAGTCCAGTCAC). In addition, we manually double-checked results on miR-30b/c sequencing (considering raw reads before stripping the adapter). We confirmed the modification observed for miR-30b/c, with adenylation as the prevalent one. We would like to point out that the original title for y-axis in **Fig. 2f** might have been misleading, since the percentage reported were relative to total miR-30c (or 30b) tailing (i.e only the 3'NT reads, which are only ~5% of the total miR-30c reads) rather than to the total miR-30c (or 30b) reads, as in **Fig. 3i** and **Fig. 4i**. We aligned the quantification of **Fig. 2f** accordingly. Furthermore, we always check quality of sequencing in the FASTQC report, both at per-base and at per-sequence level, and use experiments with a base quality >30 (Phred Score). Therefore, we can exclude that some of these tailing events are results of sequencing errors. These details have been now added to the amended Methods.

9) *From **Figure 3F**, it seems clear that there were a number of miRNAs that up/down regulated upon MRE-KO. What are the identities of these miRNAs and were these changes related to the TDMD mechanism the authors proposed?*

10) *What does the coloring of the points in **Fig 3F** represent?*

R. We thank the reviewer for their comment. In addition to miR-30b and 30c, a number of miRNAs were also regulated in MRE-KO cells. The identity of such miRNAs was originally reported in **Supplementary Table 2**, and now also shown in **Fig. 3g** of the revised manuscript. None of these regulated miRNAs appeared related to the TDMD mechanism, as they do not show complementarity to the region of Serpine1 transcript that has been deleted. Hence, such deregulation is likely to be a secondary effect, due to clone adaptation to a condition in which Serpine1:miR-30 interaction has been disrupted. Indeed, in the reconstitution experiment only miR-30c was significantly reduced (**Fig. 4e**). The coloring in **Fig. 3g** is related to miRNA family and now it is explained in the figure legend of the revised manuscript.

11) ***Figure 4B**- 30b-5p and 30e-5p results look fairly similar but the authors labeled one as $p=0.059$ and the other as "n.s". What is the p-value for 30e-5p?*

12) ***Figure 4D** - results for 30b-5p is missing.*

R. We thank the reviewer for their comments. We have added details on p-values in **Fig. 4b** for miR-30e ($p=0.174$) and all the remainder miRNAs. The p-value has been calculated with Welch's Test on independent measurements made by RT-qPCR. Results from reconstitution experiments are shown for miR-30b-5p (as for any other miRNA) in **Fig. 4e** by small RNA sequencing, comparing WT vs. Mutated Serpine1 (SE1) constructs. The 'seed' mutant was excluded in the plot, as it was incapable of producing any effect on miR-30s, and miR-30c and 30b in particular (as shown for instance in **Fig. 4f**, where tailing was plotted). A complete set of data (including small RNA

sequencing upon seed mutant expression) was included in the GEO database as reported in **Methods**. The different behavior of miR-30c Vs. miR-30b in reconstitution experiments could be explained as the consequence of a different affinity of miR-30b Vs. miR-30c for ectopically expressed serpine1-MRE. It should be also mentioned that in reconstitution experiments miRNAs were analyzed at 36 hours post infection, thus it is possible that more time is needed to observe significant reduction for miR-30b, as it is expressed at relatively higher levels than miR-30c. We have made these conclusions clearer in the revised manuscript (page 15, lines 345-349).

13) **Figure 4F**. Tailing results for 30a/d/e are missing.

14) **Figure 5D**. The color of the lines (red + blue) differ from the labels which are in green and gray.

R. We thank the reviewer for their comments. **Fig. 4** and **Fig. 5** have been amended accordingly.

Reviewer #2 (Remarks to the Author):

The Target-Dependent miRNA Degradation (TDMD) is a relatively recently described phenomenon. Its principles were established with the use of artificial targets and it remained unknown whether TDMD indeed operates in more physiological settings and is induced by natural miRNA targets. Ghini et al. provide strong evidence that Serpin1 mRNA may act as TDMD inducer. Several lines of evidence support their conclusions. The work is generally carefully done with the use of state-of-the-art assays, also providing indications of the biological effect of miR-30b/c degradation on apoptosis and cell cycle. Identification of the natural inducer of TDMD is certainly a very important finding which will greatly stimulate research in the field.

A few problems, however still remain and they will need to be addressed before the paper is considered for publication.

Major points:

2.1) *Although evidence that miR-30b/c but not other family members (miR-30a/d/e) are targeted for degradation by Serpine1 is quite strong and strongly suggests that extensive complementarity at the miRNA 3'-end is required for degradation to occur, I find the effect of about 2-fold decrease of miR-30b/c on endogenous targets and cell behavior rather surprising. Based on cpm data in **Fig. 2C** (number of RNA Seq reads), miR-30b/c makes only ~10% of the total amount of the family. Hence, after TDMD of miR-30b/c, 95% of the total complement of the miR-30 family remains intact (all family members share the seed). Why then such a strong effect on endogenous targets and on cells. The authors do not discuss this issue at all. Are miR-30 variants with less of the 3' complementarity so much less effective in miRNA repression? This could be addressed experimentally by testing them individually with one or two artificial reporters.*

R. We thank the reviewer for raising these important points, which were similarly pinpointed also by other reviewers (reviewer#1, point 1.2; reviewer #3, points 3.1 and 3.2). As miRNA sequencing is not perfectly quantitative in comparing levels of different miRNAs, we have carefully quantified all the members of the miR-30 family by absolute quantification (**Fig. 3a**), checking the cross reactivity of qPCR assays (qiagen) for miR-30s (**Supplementary Fig. 3b**). MiR-30c levels were independently verified by Droplet Digital PCR (ddPCR, **Supplementary Fig. 3d**), which uses a different qPCR assay (Exiqon). Thus, we can conclude that in 3T9 fibroblasts approx. 40% of total miR-30s is made up of 30b/c members (shown now in **Fig. 3a**). We also considered the degree of target specificity of miR-30b/c as opposed to general miR-30s targets. As stated previously (reviewer#1, point 1.2), two independent studies mapping miRNA:target interaction by CLASH (Helkaw A et al. Cell 2013 – David Tollervey Lab) and CLEAR-CLIP Seq (Moore MJ et al. Nat Comm 2015 – Bob Darnell lab) suggested that miR-30b/c preferentially binds those targets that possess a supplementary pairing in the 3' region, and that Serpine1 appears to be one of their

preferred targets (n.2 out of 935 preferred miR-30c interactions – Moore MJ et al. Nat Comm 2015). Besides, we used CLEAR-CLIP data to gain an insight into the regulation of miR-30 targets upon TDMD disruption (included in **Fig. 5h,i**) and to provide evidence in favor of a certain degree of specificity towards miR-30b/c targets (**Fig. 5j,k**). We have also to point out that the magnitude of effects is much higher during serum stimulation, when in absence of Serpine1 mediated degradation, miR-30b/c levels remain high and exert a significant repression on their targets (median repression is $-0.20 \log_2FC$), as opposed to a steady-state condition (growing cells, where the median repression is $-0.05 \log_2FC$). Such difference is now clarified in **Figs 5-6-7**, by comparing growing and serum-stimulation analyses side-by-side. We have now included and discussed thoroughly such relevant information in the revised manuscript (page **8**, lines **163-165**, and page **11**, lines **244-246** and page **12**, lines **247-262**).

2.2.) *It is not clear, particularly for the second part or the results, starting with **Fig 3** and further, under what conditions of cell growth are all the assays and quantifications done. Are the authors always looking at cells treated with serum to induce Serpine1 expression? If so, at what hr post-induction? In cells grown under control conditions, the level of Serpin1 mRNA is very low (see, for example, **Fig. 4c**). At this level of Serpin1 mRNA, one would not expect a particularly strong effect on the miR30c/b level (despite the fact that small ~2-fold difference is observed in corresponding figure). Hence, the difference between WT and MRE-KO cells shown in **Fig. 5b** (5.5 versus 7.7-fold repression of the reporter) is rather surprising. Same question applies to other panels in **Fig. 5** and the following figures. The aforementioned confusion extends to discussion. On p. 15, line 9 bottom, it is said that target mRNA (Serpine) and miR30b/c are present at ~200 cpc. This is not consistent with data in **Fig. 4c/d**.*

R. We thank the reviewer for their comment. We agree that we have to better specify conditions under which the analyses were performed. We updated the manuscript with revised figures. In particular, in **fig. 3, 5, 6** and **7** we have now distinguished between the analyses performed in growing conditions (steady-state) and those performed during serum stimulation (dynamic conditions). With regard to the quantification of miR-30b/c and Serpine1, we have now performed absolute quantification of all miR-30s (**Fig. 3a**), checked the cross reactivity of qPCR assays (qiagen) (**Supplementary Fig. 3b**) and independently verified miR-30c levels by Droplet Digital PCR (ddPCR, **Supplementary Fig. 3d**). In growing conditions, Serpine1 is expressed at relatively low levels (~ 100 cpc) but in range with miR-30c (about 70 cpc). As stated previously (R.1.4 – reviewer #1), in order to induce significant miRNA degradation, the target does not need to reach the effective abundance of all miRNA target sites, as every time the miRNA interacts with the target it is irreversibly subtracted from the miRNA pool by degradation ('multiple turnover activity'; de la Mata, et al., EMBO Rep 2015). The level of miRNA will decrease to reach a new equilibrium, where the miRNA levels are lower and degradation intermediates are increased (*Steady-state levels*). During serum stimulation, Serpine1 is expressed at extremely high levels (up 4000 cpc) for a few hours and miR-30c degradation is thus accelerated in a precise time frame (*dynamic response*). As previously stated, we have updated the discussion by including comments on the quantitative requirements of TDMD (page **15**, lines **325-342**).

2.3.) *Previous studies reported quite different requirements for the 3' end complementarity in order to induce TDMD. While Ameres et al. found that even 5-6 non-complimentary 3'-terminal pairs are tolerated, de la Mata et al. found very strong effect of even two mismatches. The authors' data suggest that the miR-30 /Serpine TDMD tolerates at least 4 mismatches. This issue needs to be discussed.*

R. We thank the reviewer for their comment. As stated previously (R1 of reviewer #1), with regard to ectopic expression, pairing requirements at the miRNA 3' end were largely analyzed in previous papers (Ameres et al. Science 2010, De la Mata et al EMBO Reports 2015), leading to the conclusion that i) extensive but not complete pairing is required and ii) reducing the pairing (even 1 nucleotide) strongly impacts on TDMD effectiveness. In agreement with these conclusions, we showed that the different levels of complementarity between Serpine1 and the five members of

miR-30 family resulted in different degrees of sensitivity to target-induced degradation. The most potent TDMD effect was observed for miR-30c, which has a match of 11 (10 consecutive) nucleotides (**Fig. 2a**, originally indicating a match of 9 nt due to a typo, has been amended), while miR-30b (which has two additional 3' mismatches) was still affected, even if to a minor extent (see **Fig. 3g**). The preference for miR-30c became more evident when, in the reconstitution experiments, Serpine1 MRE had been ectopically expressed (see **Fig. 4b,e**). Differently, lacking of extensive complementarity, miR-30a/d/e were insensitive to Serpine1 induced degradation. To further characterize Serpine1 pairing requirements, we generated a set of mutants with different levels of complementarity in the 3' region (see **Fig. 4g,h**). Our results showed that pairing of 8 nucleotides is needed for TDMD to occur and that mismatches can be tolerated if they are close to the bulge, while strongly impairing miRNA degradation when located in proximity of miRNA 3' end. This new piece of information has now been included in the revised manuscript (page **11**, lines **223-227**, and page **14**, lines **322-324**).

2.4.) *The authors investigated before, globally, turnover of all miRNAs expressed in fibroblasts, putting them into different decay categories. To what category were the miR-30 miRNAs classified?*

R. We thank the reviewer their comment. The decay rates of miR-30s were calculated by the 'pulse-chase' method in 3T9 cells (under growing conditions). Two miRNAs, 30a and 30e were found to have very low turnover rates (half-life greater than 24 hours and, thus, classified as 'slow decay' miRNAs). Conversely, miR-30b, 30c and 30d showed faster turnover rates (half-life between 12 and 24 hours, thus, classified as 'intermediate decay' miRNAs). The actual half-life of 30b was 12.5 hours, while miR-30c was 16.20 hours (R-square of the fitting=0.668). Hence, in steady-state conditions (growing cells) miR-30b/c are relatively unstable, and certainly less stable than cognate miR-30s (30a/e in particular). When Serpine1 was massively induced upon serum stimulation, the half-life of miR-30b/c decreased to 4-6 hours, as, within 4 hours, cells approximately halved the levels of these two miRNAs (-0.88/1.13 log₂ fold change) in absence of major changes in transcription and processing (see **Fig. 2c**). Unfortunately, it is not possible to exploit the 'pulse-chase' method to confirm this conclusion and exactly calculate half-lives during serum stimulation, as 4sU incorporation rate is not constant over time due to the cell metabolism being activated by serum. We have updated the discussion of the revised manuscript to include this conclusion (page **15**, lines **330-331** and **334-335**).

2.5.) *In Fig. 4c/d, expression of the seed mutant is very low, perhaps too low to see any effect. Why only single MOI is used for this important mutant? Could higher MOI be tested to make this control stronger?*

R. We agree with the reviewers (reviewer #1 also mentioned the same issue). We performed additional experiments to express the 'seed' mutant at higher doses, in line with WT and MUT constructs. Results confirm that the seed-match is not sufficient for degradation even at extremely high doses (>10.000 cpc). We amended figures accordingly (**Fig. 4c,d**).

2.6.) *In Fig. 4E, in contrary to the text, the level of miR-306 is changed. This issue and potential explanation is discussed in Discussion. This observation needs also to be mentioned on p.10 to avoid confusion.*

R. We hypothesize a typo and assume the reviewer is mentioning the different behavior of miR-30c and 30b in reconstitution experiments. As stated in the discussion, such behavior could be the consequence of a different affinity of Serpine1 for miR-30b/c upon ectopic expression. It should be also mentioned that in reconstitution experiments miRNAs were analyzed at 36 hours post infection, thus it is possible that more time is needed to observe significant reduction for miR-30b, as it is expressed at relatively higher levels than miR-30c. We have made these conclusions clearer in the revised manuscript (page **15**, lines **21-23**).

2.7. Figure 4G. *It is not clear how the levels of SEI WT reporter were varied to obtain the presented correlation. A battery of different MOIs was used?*

R. The correlation was generated by calculating the copies of Serpine1-MRE (wild type) and miR-30c-5p in independent experiments with different MOIs. Following indications from reviewer #3, we removed this panel from the revised manuscript as it was not providing any significant conclusion.

Minor points:

1. P. 10, l. 9 top. Rather: "... Serpine 1 constructs" (plural)
2. P. 7, l. 3 bottom. Rather "... remained ALMOSTunchanged...." Some effects are seen
3. P. 11, l. 5 top. Sed (?) fluorescence?
4. P. 16, l. 6 bottom. Referring to ceRNA-like effect is not a good idea. CeRNA was never documented to involve miRNA degradation.

R. We thank the reviewer for their comments. We have amended the manuscript accordingly.

Reviewer #3 (Remarks to the Author):

RNAs can induce degradation of miRNAs if they bind them with extensive complementarity in a process also known as TDMD. However, endogenous TDMD-inducing RNAs have remained elusive. Ghini et al. identify Serpine1 mRNA as one such RNA, acting on miR-30b & c. They further report that this regulation is biologically relevant in serum-stimulated cells, where failure to reduce miR-30b, c levels causes inappropriate target silencing, and ultimately cell death.

I find the identification and characterization of Serpine1 as a TDMD-inducer convincing and interesting. I am less convinced by the deduced biological implications as detailed below. The authors should address the following issues:

Major points

3.1.) *The authors claim that Serpine1 mRNA specifically initiates TDMD on miR-30b/c but not three other miR-30 family members. Loss of this Serpine1-mediated TDMD (through MRE knock-out) causes specifically upregulation of these two family members, which then results in increased silencing of miR-30 family targets, and cell death. This scenario is unconvincing, because miR-30b and c appear to account only for a very small fraction of total miR-30 family miRNA molecules in the cell (cf. **Fig. 2C**). This could be a consequence of sequencing biases, which should be ruled out by absolute quantification of all family members. However, if, as likely, true, it is unclear how such a modest change of overall miR-30 level would then decrease target levels. Indeed, although the data indicate a ratio of one for miR-30c to Serpine1 mRNA levels (**p. 8, para 2**), this ratio would be much diminished by competing Serpine1 binding through other family members.*

R. We thank the reviewer for their comment, which was raised by other reviewers (reviewer #1 point 2 and reviewer #2 points 1 and 2). As far as the quantification of miR-30b/c and other miR-30s is concerned, we have now performed absolute quantification of all miR-30s (**Fig. 3a**), checked the cross reactivity of qPCR assays (Qiagen) (**Supplementary Fig. 3b**) and verified miR-30c levels by Droplet Digital PCR (ddPCR, **Supplementary Fig. 3d**). We can thus conclude that in 3T9 fibroblasts, a significant fraction (approx. 40%) of total miR-30s is made up of the 30b/c members (shown now in **Fig. 3a**). Following this quantification, we updated the Target:miRNA ratio in **Fig. 3c**. With regard to the interaction between Serpine1 and miR-30c and a possible competition with other miR-30s, we have looked at two independent studies mapping miRNA:target interaction by CLASH (Helkawa A et al. Cell 2013 – David Tollervey Lab) and CLEAR-CLIP Seq (Moore MJ et

al. Nat Comm 2015 – Bob Darnell lab). Both studies suggested that miR-30b/c preferentially binds those targets that possess a supplementary pairing in the 3' region, and that Serpine1 appears to be one of the preferred targets (n.2 out of 935 preferred miR-30c interactions – Moore MJ et al. Nat Comm 2015). Finally, we used CLEAR-CLIP data to gain an insight into the regulation of miR-30 targets upon TDMD disruption (included in **Fig. 5h,i**) and to provide evidence in favor of a certain degree of specificity towards miR-30b/c targets (**Fig. 5j,k**). This new set of data have now been included and discussed in the manuscript (page **8**, lines **163-165**, and page **11**, lines **244-246** and page **12**, lines **247-262**).

3.2. *Consistent with the concern about miRNA family member levels, although **Fig. 5D** shows a statistically significant increase in target silencing, the differences in predicted target level repression in MRE-KO vs. wt appear minute (5%). Even with the modest effects we have come to accept in the miRNA field, it is hard to imagine how this would result in a biological effect. Authors should show directly that a <2-fold over-expression of miR30b/c (during continued expression of other miR-30 family members) recapitulates the effects of MRE-ko on both targets and cell death/cell cycle (below).*

R. We thank the reviewer for their comment. Indeed, in growing conditions the differences in predicted target level repression in MRE-KO vs. WT cells are of minor intensity (5%), although significant. This is a steady-state in which mutant cells are adapted to the loss of Serpine1:miR-30b/c interaction and to the new (increased) levels of miR-30s. As consequence, 244 transcripts are significantly changed, with no enrichment for miR-30 targets, suggesting that such differences are largely secondary (or indirect) effects (**Fig. 7a,b**). Nevertheless, these effects appear to be specific, as they are found in both MRE clones, and may explain in part the biological differences observed in acute conditions (i.e. sensitivity to cell death). To gain a further insight in primary effects following TDMD disruption, we thoroughly analyzed gene expression upon serum stimulation. This is a more dynamic condition characterized by the rapid downregulation of miR-30b/c, which is abrogated in mutant cells. Effects on miR-30 targets are much higher in this setting. Targets are repressed to approx. 20% (**Fig. 5b,c,e,g,i**) and a large number of transcripts are significantly changed (**Fig. 7a,c**), with enrichment for miR-30 targets (**Fig. 7d,e**). A certain degree of specificity towards miR-30b/c preferential targets (as opposed to more general miR-30 targets) was observed under these conditions (**Fig. 5j,k**). This new piece of information has now been included in the revised manuscript (page **11**, lines **24-26** and page **12**, lines **1-16**). We also thank the reviewer for suggesting a relevant experiment. A mild overexpression of miR-30c would permit to identify those targets and those effects that are specific for miR-30c upregulation. However, there are some concerns about this approach. Obtaining a homogenous and mild overexpression of miR-30c is technically challenging, as a simple transfection at low doses of miR-30c mimic would result in high cell-to-cell variability, with some cells remaining untransfected and some expressing miR-30c at high doses. Such experiment would also fall out of the scope of this manuscript, which is about the involvement of endogenous targets in TDMD and aims at providing evidence that TDMD affects miRNA activity. We believe that the new set of experiments and analyses that has been added (now shown in **Fig. 5, 6** and **7**) strongly support this conclusions. In addition, we have amended the manuscript to distinguish between primary (miRNA upregulation, change in miRNA isoforms, target downregulation) and secondary effects (change in gene expression, cell behavior) of Serpine1:miR-30b/c interaction, the two being deeply intertwined.

3.3. *In the same vein, the 'miR-sensor' used in **Fig. 5B** contains 4 sites of perfect complementarity to miR-30c. This would immediately bias the read-out to miR-30c, away from other family members, although the sensor is claimed (p.10, last para) to reflect miR-30 (family) activity. I would agree that sensing changes of the entire miR-30 family is what needs to be done when testing the authors' model, but for this they should use a seed match, not a perfect match, target reporter. Alternatively, if the authors have identified endogenous targets of miR-30c that are relevant for the phenotypes, and specifically only regulated by miR-30c but not by other family members, they should investigate these, endogenously or at the level of a derived reporter. This would reflect a*

different hypothesis from that stated on p. 10 though. Incidentally, repression did not increase by 7.7-fold but to 7.7-fold in the MRE-KO (description of Fig. 5B, p.11, para 1).

R. We agree with the reviewer. The sensor that has been used to measure miRNA activity is designed to follow miR-30c activity and not miR-30 family activity. This has been amended in the revised manuscript (**page 11, lines 11-14**). Indeed, miR-30c (and 30b) appears to preferentially bind to targets that possess a supplementary pairing in the 3' region. CLEAR-CLIP data were used to gain an insight into the regulation of specific miR-30b/c targets upon TDMD disruption (included in **Fig. 5** and in **Fig. 7**). In our view, regulation on multiple endogenous targets, either predicted by TargetScan or isolated through biochemical approaches (HITS-CLIP or CLEAR-CLIP) is more relevant than a synthetic sensor or an individual reporter. Hence, miR-sensor data were moved to **Supplementary Fig. 6** and new analyses on endogenous targets were included in **Fig. 5** and **Fig. 7**. Based on the reviewer's suggestion, we have also discussed some of the miR-30c targets (i.e. Bcl6, Sirt1, Ccne2, Per2, Cry2) that are relevant for our model and that might explain the effects observed on cell behavior (**page 13, lines 295-298** and **page 14, line 299**).

3.4. In Fig. 6B, the authors imply that "morbidity or mortality", "organismal death" and "perinatal death" are all aspects of "cell death" – I consider this a leap and most likely wrong, since they all refer to organismal not cellular level. Accordingly, although MRE-KO cells appear to be more prone to cell death under stress conditions, the evidence that this is through "over-repression" of miR-30b/c targets is weak. Again, recapitulation of these phenotypes by modest miR-30b/c over-expression (point #2 above) would at least provide some circumstantial support for the model, although ideally there would be some data to implicate a specific set of targets.

R. We thank the reviewer for their comment. We revised the functional characterization of genes in mutant cells. This has now been included in **Fig. 7**, comparing side-by-side effects in steady-state conditions (growing cells) and those during serum stimulation of quiescent cells (a more dynamic condition). As stated previously, we believe that 'MRE-KO cells appear to be more prone to cell death under stress conditions' as a consequence of both primary and secondary effects that originate from the disruption of Serpine1 mediated degradation of miR-30b/c. A number of regulated targets that might explain these effects are discussed (**page 13, lines 295-298** and **page 14, line 299**).

Additional points:

1. Fig. 1E, p.7, para 1: Please define the "max contribution of each single target to the total pool of 3C-targets of a given miRNA" – is this based on number of predicted targets overall, above the expression cutoff as defined in Fig. 1D, or weighted by abundance? I assume the latter, but it is not clear. A key to convert bubble size to target abundance would also be helpful.

2. To validate that miR-30b/c expression level changes occur at the level of the guide strand, please provide from the sequencing data a plot of passenger strands for all five family members over time

3. Fig. 3D/E: Serpine mRNA and protein levels appear unchanged or even down in the MRE mutant. This seems surprising because it ought to be a target of the whole miR-30 family so that loss of MREs would be expected to cause upregulation. Please explain.

4. Fig. 4D,E: Why does miR-30b-5p remain unchanged by reintroduction of wt-Serpine, and what would the corresponding graph for Fig. 4D look for it?

5. Fig. 4F: Why are n's all different for the three transgenes, have data been omitted? If so, please justify.

6. Fig. 4F vs. 3H: tailing levels appear globally reduced (irrespective of the type of transgene added) in 4F vs. what is reported for the ko cells in 3H. This difference is much bigger than the effect that results from introducing wt-Serpine1, making me wonder how robust this assay is – please comment.

7. **P.10, para 2/ Fig. 4G/H:** *The authors argue that they require much higher levels of Serpine1 mRNA to elicit degradation when transfected than when endogenously present because there is only modest transduction efficiency at the low value. I can accept this argument. However, Ghini et al. then use the same experiment to argue that Serpine1 mRNA has a dose-dependent, not a thresholded, function on miR-30c-5p levels. Clearly, if the preceding explanation were true, this experiment would be unsuitable to provide evidence for or against a threshold mechanism, so I cannot accept their conclusion.*

R. We thank the reviewer for the comments.

1) we calculated the max contribution by weighting predicted targets by abundance. We made the text clearer and added the legend for bubble size.

2) we checked miR-passenger expression (which is unchanged) and provided a plot for each passenger of miR-30s during serum stimulation in **Supplementary Fig. 2b**. Expression data for passenger miR-30s in mutant cells are in **Supplementary Table 2**.

3) Protein levels of Serpine1 appear unchanged, which is somehow surprising since it is a specific target of miR-30b/c and it is expected to be increased in mutant cells. Indirect effects due to adaptation of mutant clones could explain this behavior. We noticed that miR-199a-5p and miR-145-5p, which both target Serpine1, were upregulated in mutant cells (see **Fig. 3g** and **Supplementary Table 2**) and could thus maintain physiological levels (or even slightly downmodulating) of Serpine1 protein.

4) The different behavior of miR-30c and 30b in reconstitution experiments could be explained by the different levels of target complementarity of their 3' regions, a feature which strongly affects TDMD. This would be consistent also with data obtained from Serpine1 mutants and shown in **Fig. 3g,h**. It should also be mentioned that in reconstitution experiments miRNAs were analyzed 36 hours post infection, thus it is possible that more time is needed to observe significant reduction for miR-30b, as it is expressed at relatively higher levels than miR-30c and has a lower affinity for Serpine1. We have made these conclusions clearer in the revised manuscript (page 15, lines 345-347).

5) No data were omitted. This was the original set of independent biological samples that were analyzed by small RNA sequencing (total of 12 samples) and expressed at similar MOI. We amended Fig. 4e, including the correct sample numbers.

6) MiR-30b/c tailing reads approximately represent 5% of total reads and half of them are adenylation variants. **Fig. 4f** (rescue experiments) shows tailing events in mutant cells upon reconstitution, divided into A-forms, U-forms and A/U mixed. In this setting, mutant cells (MRE-KO) showed basal levels of adenylation (~1% for miR-30b and ~0.5% for miR-30c) that increased up to 2-3% upon the re-expression of wild-type form of Serpine1 MRE. As shown in **Fig. 3i** (mutant Vs. wild-type cells), A-forms in mutant cells were ~1% in the case of miR-30b and ~0.5% in the case of miR-30c. In parental cells these values at least doubled. Therefore, adenylation data are consistent. We agree with the reviewer that % of U-forms (which are not regulated and do not correlate with TDMD) is different in the two figures relating to miR-30c. However, it should be pointed out that uridylation of miR-30c might be challenging to assess as mono-uridylation (+1 U variant) is formally identical to templated isomiR (i.e. miR-30c-1 and miR-30c-2 pre-miRNA have a Uridine as first base at 3'end of the canonical miR-30c-5p sequence: 5'uguaaacaucacuacacucagc3'-Ugu).

7) We completely agree with the reviewer. The two conclusions cannot stand together. We, thus, removed, comments and panels related to threshold mechanism, as we cannot prove a dose-dependent effect.

Minor points

1. *Phrasing tends to be at times vague or misleading, please carefully review. A few examples are below:*

a) *p.3, para1: “...however, other sequences, such as compensatory sites located...” – as it is not clear, what would be “compensated for”, the authors should use “complementary” not “compensatory”, and do so consistently throughout the manuscript. “Sites” is also misleading since the miRNA binding site would be composed of stretches of seed complementarity and complementarity to the 3’end.*

b) *ibid: “...hundreds to thousands of RNAs could interact with a single miRNA, often simultaneously...” – this reads as if one molecule of miRNA can silence many targets in parallel, but I assume what the authors want to say is that within a cell type, miRNAs can have many different targets?*

c) *ibid: “The interaction between miRNAs and targets occurs at the level of Argonaute proteins...” – what does this mean?*

d) *The authors refer to De et al. 2013, de la Mata et al. 2015, and Park et al. 2017 as “in vitro” studies, p.4, p.6, which is a bit confusing considering that two of the studies were performed in a test tube, the third one on cells. More clarity would be useful, also in assigning an appropriate label to their own study.*

R. Agreed. We reviewed the entire manuscript and corrected the phrasing.

2. *Krol et al. Nat Rev Genet 2010 is cited as a review on TDMD, but does not treat the topic; an appropriate reference should be given. Item, Krol et al Cell 2010 does not report on TDMD.*

R. Agreed. We replaced the wrong citation with Ruegger and Groshans (Trends in Biochemical Sciences, 2012) (page 3 line 58). We also corrected the other reference (Page 4 line 68).

3. *Fig. 1A: define “NCS” and “CS” in figure legend.*

R. Agree. We defined NCS and CS in the legend **Fig.1a**.

4. *Fig. 1D: how were the thresholds for expression filtering selected?*

R. Thresholds were based on technical reproducibility of the data. Below 10 CPM, miRNA sequencing is very noisy and miRNAs are expressed at extremely low levels. Same applies to transcripts.

5. *The discussion should be more concise and focused, e.g. the question of target vs. miRNA stoichiometry is important does not need to be discussed twice (p.14, p.17).*

R. Agreed. We revised and shortened discussion in the new manuscript.

REVIEWERS' COMMENTS:

Reviewer #1 (Remarks to the Author):

The authors have done a nice job of addressing my original concerns and comments and the revised manuscript should be published.

Reviewer #2 (Remarks to the Author):

I am satisfied with responses to my comments, and also to those of other referees. The manuscript could be published in this revised form.

Reviewer #3 (Remarks to the Author):

The authors have adequately addressed the major points that I raised. Before proceeding to publication, the following issues should be considered:

- This is a nice story, but the presentation is sub-optimal. Writing would benefit greatly from careful proofreading for clarity, grammar, and appropriate terminology. For instance: ll.39-40 "targets are bound through a Watson-Crick base pairing", l. 50 "seed match efficacy", l.72 "ectopically expressed artificial targets" (how could they not be ectopic?), ll.101-103 ".. of all the interactions" (=> predictions), ll. 165-167 & 249-251 - ???, ll-292-295 - "anticipated upregulation", ll.325-328 - "quantitative factors", "absolute approaches"??
- I understand that de la Mata et al., EMBO Rep (2015) defined TDMD as "target-directed miRNA degradation", not "target-dependent miRNA degradation" - the authors might want to avoid confusion by sticking to the original term in title and text.
- Some claims in abstract and discussion appear unnecessarily inflated; this is not needed for readers to appreciate the importance and novelty of the results by Ghini et al. For instance, (ll.23-25) "endogenous targets that induce miRNA degradation remain elusive" should be toned down considering the recent work by Bitetti et al., NSMB 2018. The tone of discussion of that paper (ll. 363-365) also appears unnecessarily antagonistic (perhaps unintentionally so) - the effects shown by Bitetti et al. appear quite clear to me. This does not diminish the importance of the work by Ghini et al. Finally, the 2-fold degradation of miR-30b/c by TDMD seems very much in line with the observations of de la Mata et al., EMBO Rep (2015) on TDMD function in fibroblasts, I do not understand how the new results would be in contradiction (ll. 314-315).
- I still don't understand citation of ref. 17 (l. 64) in the context of TDMD.

Response to Reviewers' comments:

Reviewer #3 (Remarks to the Author):

The authors have adequately addressed the major points that I raised. Before proceeding to publication, the following issues should be considered:

1. This is a nice story, but the presentation is sub-optimal. Writing would benefit greatly from careful proofreading for clarity, grammar, and appropriate terminology.

R. We thank the reviewer for the comment. As requested we carefully checked the manuscript with the help of a scientific author's editor. The English language has been now improved through all the manuscript.

In particular, reviewer examples has been amended as following:

ll.39-40 “targets are bound through a Watson-Crick base pairing”,

now is “Targets are bound through base pairing between the miRNA and their miRNA responsive elements (MREs)”

l. 50 “seed match efficacy”

now is “seed type hierarchy”

l.72 “ectopically expressed artificial targets”

now is just “artificial targets”

ll.101-103 “.. of all the interactions” (= > predictions),

now is “the predicted interactions”

ll. 165-167 & 249-251 – ???,

the sentences have been rephrased

“miR-30c and Serpine1 were expressed at comparable levels (Serpine1: 170 cpc in GW cells and 88 cpc in quiescent cells; Fig. 3b), with a target per miRNA (TPM) value of ~1”

“We, then, compared target repression in MRE-KO and wild type cells.”

ll-292-295 – “anticipated upregulation”,

now is “earlier induction”

ll.325-328 – “quantitative factors”, “absolute approaches”??

now is “ context dependent quantitative requirements”

2. I understand that de la Mata et al., EMBO Rep (2015) defined TDMD as “target-directed miRNA degradation”, not “target-dependent miRNA degradation” – the authors might want to avoid confusion by sticking to the original term in title and text

R. We amended the text accordingly. “target-induced” or “target-dependent” has been now replaced in the manuscript (title, abstract and text) by “target-directed” (line 2, 23, 56, 109, 140, 186, 203 and 219).

3. *Some claims in abstract and discussion appear unnecessarily inflated; this is not needed for readers to appreciate the importance and novelty of the results by Ghini et al. For instance, (ll.23-25) “endogenous targets that induce miRNA degradation remain elusive” should be toned down considering the recent work by Bitetti et al., NSMB 2018. The tone of discussion of that paper (ll. 363-365) also appears unnecessarily antagonistic (perhaps unintentionally so) – the effects shown by Bitetti et al. appear quite clear to me. This does not diminish the importance of the work by Ghini et al. Finally, the 2-fold degradation of miR-30b/c by TDMD seems very much in line with the observations of de la Mata et al., EMBO Rep (2015) on TDMD function in fibroblasts, I do not understand how the new results would be in contradiction (ll. 314-315).*

R. We amended the abstract as suggested.

line 24 – “*endogenous targets that induce miRNA degradation remain elusive*” now is “but further investigation on endogenous targets is necessary

lines 363-364 – “*However, the molecular details of NREP mediated effects, in particular its impact on miR-29 targets, still remain elusive*” now is “However, the impact of NREP on shared miR-29 targets was not investigated”.

Lines 314-315 – the sentence “*it occurs in fibroblasts, suggesting that TDMD is not limited to neuronal cells*” has been removed.

4. *I still don’t understand citation of ref. 17 (l. 64) in the context of TDMD.*

R. Ref17 at line 64 was removed and replaced by Bitetti et al. (NSMB 2018).